# EDITVAL: BENCHMARKING DIFFUSION BASED TEXT-GUIDED IMAGE EDITING METHODS

## ABSTRACT

A plethora of text-guided image editing methods have recently been developed by leveraging the impressive capabilities of large-scale diffusion-based generative models such as Imagen and Stable Diffusion. A standardized evaluation protocol, however, does not exist to compare methods across different types of fine-grained edits. To address this gap, we introduce EDITVAL, a standardized benchmark for quantitatively evaluating text-guided image editing methods. EDITVAL consists of a curated dataset of images, a set of editable attributes for each image drawn from 13 possible edit types, and an automated evaluation pipeline that uses pre-trained vision-language models to assess the fidelity of generated images for each edit type. We use EDITVAL to benchmark 8 cutting-edge diffusion-based editing methods including SINE, Imagic and Instruct-Pix2Pix. We complement this with a large-scale human study where we show that EDITVAL's automated evaluation pipeline is strongly correlated with human-preferences for the edit types we considered. From both the human study and automated evaluation, we find that: (i) Instruct-Pix2Pix, Null-Text and SINE are the top-performing methods averaged across different edit types, however *only* Instruct-Pix2Pix and Null-Text are able to preserve original image properties; (ii) Most of the editing methods fail at edits involving spatial operations (e.g., *changing the position of an object*). (iii) There is no 'winner' method which ranks the best individually across a range of different edit types. We hope that our benchmark can pave the way to developing more reliable text-guided image editing tools in the future. We will publicly release EDITVAL, and all associated code and human-study templates to support these research directions in `https://deep-ml-research.github.io/editval/`.

## 1 INTRODUCTION

Large-scale text-to-image diffusion models such as Stable-Diffusion, Imagen and DALL-E (Rombach et al., 2022; Ho et al., 2021a; Balaji et al., 2023; Saharia et al., 2022; Ho et al., 2021b) have seen rapid advances over the last years, demonstrating impressive image generation capabilities across a wide set of domains. A highly impactful use-case of these models lies in using them to edit images via natural language prompts (Hertz et al., 2022; Kawar et al., 2023; Mokady et al., 2022; Zhang et al., 2022; Ruiz et al., 2023; Shi et al., 2023; Couairon et al., 2022; Meng et al., 2022; Brooks et al., 2023). This capability has a great number of industrial applications, including design, manufacturing and engineering, but can also be used as a tool to accelerate machine learning research. For example, a model can be prompted to generate counterfactual examples to probe its interpretability, or rare examples that are used to augment training datasets to improve a model's out-of-distribution robustness (Vendrow et al., 2023; Trabucco et al., 2023).

Evaluating diffusion based text-guided image editing models, however, is challenging due to the difficulties in measuring how faithfully a generated image obeys a requested edit. Moreover, there are broad classes of edits for which methods need to be evaluated. Typically, a CLIP image-text similarity score (Hessel et al., 2021) is used to quantify the efficacy of a given edit. However, these scores have been shown to not always be reliable (Goel et al., 2022). CLIP scores also cannot tease apart particular aspects of an edit, for example, if changing the position of a particular object leaves the rest of the image unchanged (Gokhale et al., 2023). These gaps could be addressed by using human evaluators, but this is usually not scalable and thus limits the scope of edits and datasets that can be considered. Moreover, human studies often lack a standardized protocol, making it difficult to fairly compare methods.

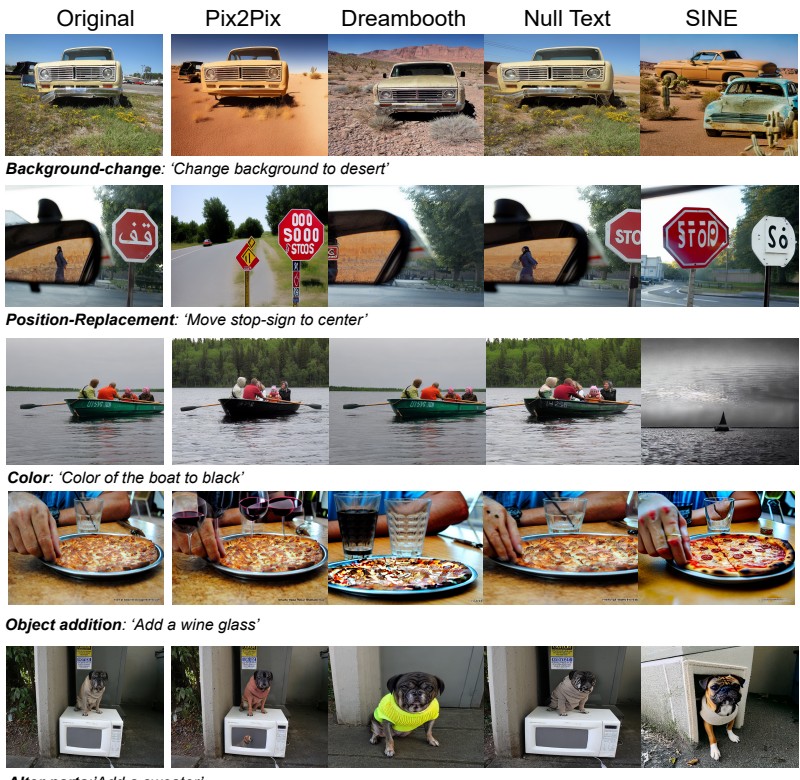

Figure 1: **Qualitative Examples from Image Editing on EDITVAL**. We find that for non-spatial edits (e.g., *Changing background, color of an object, adding objects*), Instruct-Pix2Pix performs well, while other methods struggle. For spatial edits (e.g., Position-replacement), none of the editing methods lead to effective edits.

To address these issues, we introduce EDITVAL, a standardized benchmark which can serve as a checklist for evaluating text-guided image editing methods at scale across a wide range of edit types. Our benchmark consists of 3 components: i) a curated set of test images from MS-COCO (Lin et al., 2014) spanning 19 object classes, ii) a set of manually defined editable attributes for each image based on 13 possible edit types (e.g. *adding an object, changing an object's position*), and iii) two standardized pipelines – one automated and the other a large-scale human study – to evaluate the fidelity of the edited images. Given an image and one of its editable attributes, we apply a standardized template to construct a text prompt (e.g. '*Change the position of the donuts to the left of the plate*') and give this as input to the text-guided image editing model. The generated image is then assessed using our standardized evaluation pipelines which leverages powerful pre-trained auxiliary models (e.g., object detectors) and a human study template to quantify the edit fidelity.

We use EDITVAL to evaluate 8 state-of-the-art text-guided editing methods including SINE (Zhang et al., 2022), Imagic (Kawar et al., 2023) amongst others. We first validate that the scores from automatic evaluation in EDITVAL are well-aligned with human evaluators for these models by running a large-scale human study where we find a strong positive correlation between corresponding scores. We then use EDITVAL to benchmark and probe the success and failure modes of these methods (see Fig 1 for qualitative visualizations). Overall, we find that (i) while methods such as SINE (Zhang et al., 2022), Instruct-Pix2Pix (Brooks et al., 2023) and Null-Text (Mokady et al., 2022) obtain the highest scores on EDITVAL amongst other methods, only Instruct-Pix2Pix and Null-Text are able to preserve original image properties, (ii) there is no 'winner' method which performs the best across all 13 edit types; and (iii) on complex editing operations involving spatial manipulation such as altering the position of an existing object or adding a new object at a particular position, all methods perform poorly.

We hope that our results can pave the way to developing more reliable text-guided image editing tools in the future. To our knowledge, this is the first work to compare text-guided image editing methods in a standardized manner. We, therefore, release EDITVAL, including all images, edit operations,

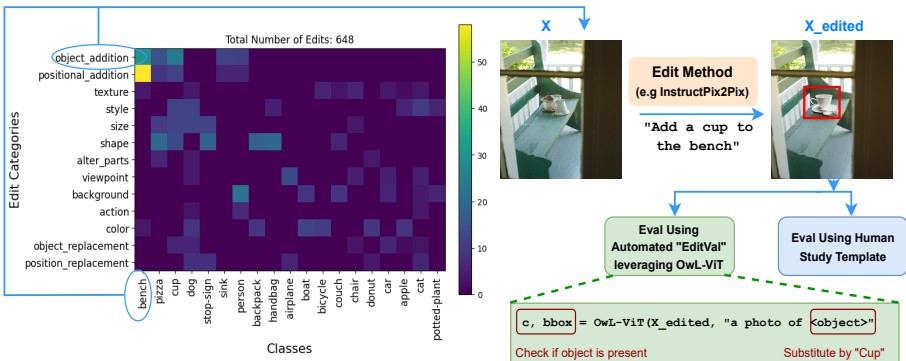

Figure 2: **EDITVAL contains 648 unique image-edit operations across 19 classes from MS-COCO spanning a variety of real-world edits.** Edit types span simple categories like adding or replacing an object to more complex ones such as changing an action, viewpoint or replacing the position of an object.

evaluation scripts, and human study which in conjunction can serve as a checklist for evaluating text-guided editing methods.

In summary, our contributions are:

- EDITVAL, a standardized benchmark dataset for evaluating text-guided image editing methods across diverse edit types, validated through a large-scale human study.
- An automated evaluation pipeline and standardized human-study template which can be used to compare text-guided image editing methods at scale.
- A comprehensive evaluation of 8 state-of-the-art image editing methods on EDITVAL. To the best of our knowledge, this is the first work to compare a large number of text-guided image editing methods at scale on a common benchmark.

## 2 RELATED WORKS

**Text-Guided Image Editing Methods**. Recently, text-guided image diffusion models (Rombach et al., 2022; Balaji et al., 2023; Ho et al., 2021b; Saharia et al., 2022; Ho et al., 2021a) have demonstrated strong image generation capabilities which have resulted in state-of-the-art FID scores on generation benchmarks such as MS-COCO. These models are usually pre-trained on a large corpus of image-text pairs such as LAION (Schuhmann et al., 2022) using a diffusion objective. Recently these powerful text-guided image generation models have been used to edit real-images(Hertz et al., 2022; Kawar et al., 2023; Mokady et al., 2022; Zhang et al., 2022; Ruiz et al., 2023; Shi et al., 2023; Couairon et al., 2022; Meng et al., 2022; Brooks et al., 2023; Epstein et al., 2023). In our paper, we evaluate all these methods except (Epstein et al., 2023) due to unavailability of their codebase.

**Image Editing Benchmarks**. To date, TedBench (Kawar et al., 2023) and EditBench (Wang et al., 2023) have been proposed as text-guided image editing benchmarks, however, both have limitations. TedBench is relatively small, evaluating on 100 images encompassing *only* highly common edit types like object addition and color changes. It also lacks evaluation of recent popular methods like SINE (Zhang et al., 2022) and Pix2Pix (Brooks et al., 2023). EditBench, on the other hand, is limited to evaluating mask-guided image editing methods which require an additional mask to be provided along with the edit prompt. Our proposed EDITVAL, instead, can be applied to any text-guided editing method including mask-guided methods. Further details comparing EDITVAL to EditBench can be found in Appendix L.

## 3 EDITVAL: EVALUATION BENCHMARK FOR TEXT-GUIDED IMAGE EDITING

Our text-guided image editing benchmark, EDITVAL, comprises three components: (i) A seed dataset $\mathcal{D}$ with carefully chosen images from $\mathcal{C}$ classes in MS-COCO; (ii) an edit type suite $\mathcal{A}$ containing different edit operations to be applied to the images in $\mathcal{D}$; and (iii) two evaluation procedures to assess the quality of the edited versions of the images in $\mathcal{D}$ for a given image editing method: one involving a human study and the other utilizing an automated pipeline with powerful pre-trained vision-language models. These components in conjunction serve as a checklist to evaluate text-guided image editing methods.

## 3.1 DATASET DESCRIPTION AND EDIT TYPE SUITE

**Selecting Edit-Types and Object Classes**. We begin by defining a set of 13 distinct edit types denoted as the edit type suite $\mathcal{A} = \{a_i\}_{i=1}^{13}$, including (i) `object-addition`, (ii) `object-replacement`, (iii) `positional addition`, (iv) `size`, (v) `position-replacement`, (vi) `alter-parts`, (vii) `background`, (viii) `texture`, (ix) `style`, (xi) `color`, (x) `shape`, (xii) `action`, and (xiii) `viewpoint` edits. Each of these edits are defined in detail in Appendix B.2. For each edit type, we employ ChatGPT to identify classes from MS-COCO for which that edit type makes sense in real-world scenarios. We motivate our choice of MS-COCO as a dataset in Appendix B.1. Specifically, we prompt ChatGPT with "*List the classes in MS-COCO for which $a_i$ is plausible*" where $a_i \in \mathcal{A}$. We validated these classes in a small-scale human-study where we ask human participants to rate if the output classes can be used in practice for incorporating the given edit-type (see Appendix H).

**Filtering Classes.** The resulting list of filtered classes across all edit types is extensive. Recognizing the time-intensive nature of most text-guided editing methods (due to fine-tuning step on each individual image), we create a dataset that, while not excessively large, is of reasonable size for validating diverse edit operations. To achieve this, human raters are engaged to further refine the list of classes, maximizing overlap among different edit types. This human-in-the-loop filtering process results in the selection of 19 classes, strategically chosen to ensure strong overlap among classes across various edit types.

**Selecting the Edit-Operations**. We curate 92 images across these 19 classes for editing, denoted as $\mathcal{D} = \{x_j\}_{j=1}^{92}$. For each edit type $a_i \in \mathcal{A}$ and object class $c_k \in \mathcal{C}$, we generate specific prompts using ChatGPT[1] to obtain the changes that are plausible for that edit type and object class. For instance, for class $c_k = $ "Bench" and edit type $a_i = $ `object-addition`, we prompt ChatGPT with "*What objects can be added to a Bench*?". This results in a unique set of edit operations for each class in $\mathcal{C}$ and each edit type in $\mathcal{A}$, which we use to construct the benchmark. After this careful curation of edit types and their corresponding edit operations, EDITVAL contains 648 unique operations encompassing a wide range of real-world image manipulations. We include this an easy-to-use json file in the following format: {class : {image-id : {edit-type : $[e_1, e_2, ..e_n]$}}}, where $[e_1, e_2, ...e_n]$ correspond to the edits to be made for the given edit type. From this, prompts can be generated in a standardized way to generate the edited image. We provide details on how prompts across different methods are selected in Table 3. Qualitative examples of the edit operations are provided in Appendix E.

## 3.2 EVALUATION PIPELINES

The third component of EDITVAL is a pair of complementary evaluation pipelines: (i) the design of a large-scale human study with accompanying standardized templates, and (ii) an automatic evaluation pipeline which leverages powerful pre-trained vision-language models to evaluate the generated image edits. We use both pipelines to assess the robustness of 8 state-of-the-art image-editing methods. Specifically, we use the human study to evaluate the generated image edits for all 13 edit types in $\mathcal{A}$, while the automated pipeline is used to evaluate a subset of 6 out of the 13 types - specifically, (i) `object-addition`; (ii) `object-replacement`; (iii) `positional-addition`; (iv) `size`; (v) `positional-replacement`; (vi) `alter-parts`. This decision was motivated by work which has shown that vision-language models (Radford et al., 2021) cannot reliably recognize concepts involving viewpoint or action (Gokhale et al., 2023), hence we use these models to only evaluate edit types with object-centric modifications and also our in-house experiments in Appendix K.

### 3.2.1 HUMAN STUDY DESIGN AND TEMPLATES

We conduct a large scale human study using Amazon Mechanical Turk to evaluate the robustness of a set of 8 state-of-the-art image-editing methods across the 648 edit operations. In this study, as shown in Fig 3, annotators view a source image from $\mathcal{D}$, an edit operation, and the edited image resulting from applying the text instruction using an image-editing method. Participants are then tasked with answering three questions regarding the edited image's quality. These questions, outlined in Fig 3, assess: (i) the accuracy of the specified edit in the instruction, (ii) the preservation of untargeted

---

[1]Version 3.5 is used

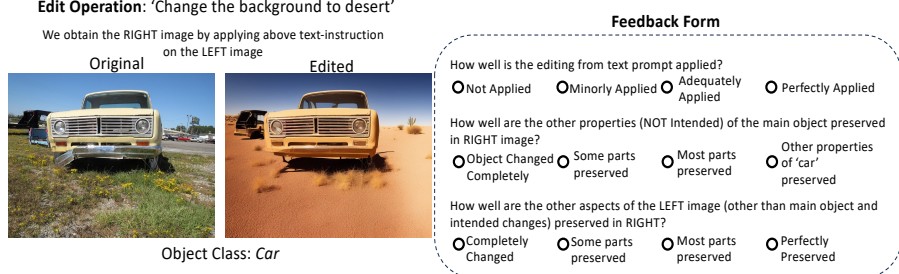

Figure 3: **Template for the AMT Human Study:** A single task displays an edit operation, a source image, an edited image from a given image-editing method and a set of questions to assess edit fidelity. Note that our human-study template does not require edited images from other methods to compare the given text-guided image editing method under evaluation (For e.g., TedBench (Kawar et al., 2023) requires edited images from all other methods). This makes our human-study template scalable and it can be independently used with *any new* editing method.

characteristics of the main object, and (iii) the preservation of untargeted parts of the image aside from the main object. For the first question, there are four selectable options ranging from the edit 'not being applied' (score: 0) to it being 'perfectly applied' (score: 3). Likewise, for the second and third questions, the options span from the characteristics being 'completely changed' to them being 'perfectly preserved.' Each level of annotation corresponds to values within the range of $\{0, 1, 2, 3\}$.

The human annotations from this study therefore enable the evaluation of image-editing methods based on (i) the success of the edit, (ii) the preservation of main object properties, and (iii) fidelity to the original image. In particular, we quantitatively measure the success of each editing method by computing the mean human-annotation score for each of the 13 edit-types (see Fig 4). We also apply several quality checks to validate the annotations, detailed in Appendix G.

### 3.2.2 AUTOMATED EVALUATION USING VISION-LANGUAGE MODELS

Given the set of edited images from any text-guided editing method, our automated evaluation procedure produces a binary score for each of the images corresponding to a subset of the edit types in $\mathcal{A}$ denoting if the edit was successful or not. Formally, given the original image $x$, the edited image $x_{edit}$, the edit type $a \in \mathcal{A}$ and one of the possible edit operations $o$ for this edit type, we define the per-image edit accuracy $R(x, x_{edit}, a, o)$ as the following:

$$R(x, x_{edit}, a, o) = \begin{cases} 1, & \text{if the edit is correct} \\ 0, & \text{otherwise} \end{cases} \tag{1}$$

CLIP (Radford et al., 2021) is effective for assessing the alignment between the edited image $x_{edit}$ and the prompt created using the edit operation $o$. However, it often fails to recognize fine-grained spatial relations (Gokhale et al., 2023) like `positional-addition`, `position-replacement`, or `size` (refer to Appendix J for a broader discussion). To address this, we instead use OwL-ViT (Minderer et al., 2022), a vision-language model with fine-grained object localization capabilities, in our pipeline. OwL-ViT is pre-trained on a vast corpus of 3.6 billion image-text pairs with a contrastive objective, and is then fine-tuned on publicly available detection datasets using a bipartite matching loss for object detection. OwL-ViT thus provides reliable bounding box annotations with object accuracies which we can leverage to validate `size`, `positional-addition` edits. We define specific rules for each edit-type in {`object-addition`, `object-replacement`, `positional-addition`, `position-replacement`, `size`, `alter-parts`} to determine whether the corresponding edit is correct. For instance, to validate an edit $R(x, x_{edit}, a, o) = 1$ where $a = $ `object-addition`, both the old object in image $x$ and the new object $o$ must be present in the edited image $x_{edit}$. We provide detailed rules for each edit operation in Appendix C. In addition to acquiring editing accuracy through Owl-ViT, we assess whether the edited image has undergone substantial changes in image context by utilizing a pre-trained DINO model. Therefore our automated evaluation consists of (i) Detector based pipeline which checks for edit-accuracy (proxy for the first question in human study) and (ii) Context preservation pipeline which checks for image fidelity and context preservation (proxy for the third question in human study).

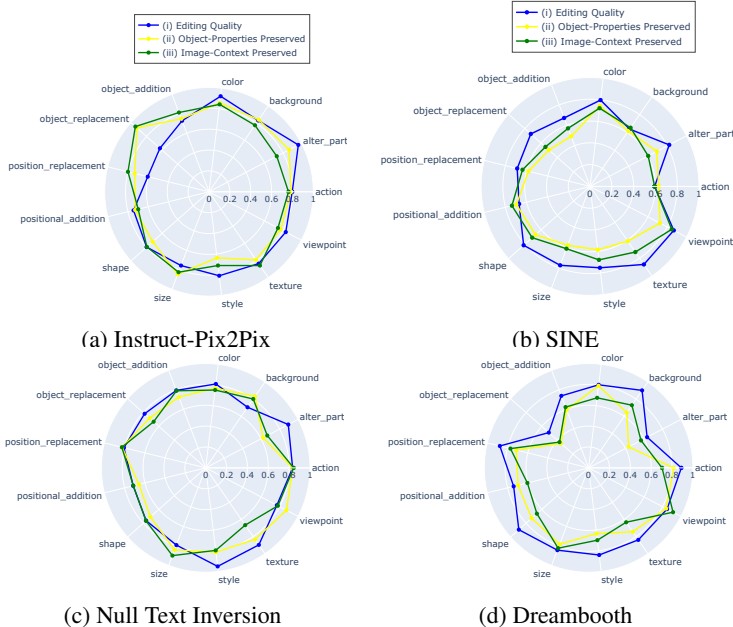

(a) Instruct-Pix2Pix

(b) SINE

(c) Null Text Inversion

(d) Dreambooth

Figure 4: **Human study results for the top 4 image-editing methods (with respect to editing accuracy) across different questions in the human study template.** (i) *Editing Quality*: We find that **Instruct-Pix2Pix, SINE, Null-Text, and Dreambooth** are the top-performing methods. (ii) *Object-Properties Preserved:* **Instruct-Pix2Pix** and **Null-text** fare well in preserving original object-properties; (iii) *Image-Context Preserved:* **Instruct-Pix2Pix** and **Null-Text** fare well in preserving the context of the original images.

## 4 EMPIRICAL RESULTS ON EDITVAL

### 4.1 IMPLEMENTATION DETAILS

Using EDITVAL, we evaluate eight of the recently introduced text-guided image editing methods: (i) Instruct-Pix2Pix (Brooks et al., 2023); (ii) Textual Inversion (Gal et al., 2022); (iii) SINE (Zhang et al., 2022); (iv) Imagic (Kawar et al., 2023); (v) Null-Text Inversion (Mokady et al., 2022); (vi) SDE-Edit (Meng et al., 2022); (vii) Diffedit (Couairon et al., 2022); (viii) Dreambooth (Ruiz et al., 2023). For all these methods, we use their public implementations with Stable-Diffusion (Rombach et al., 2022). For our automated evaluation we use the OwL-ViT (Minderer et al., 2022) implementation from Hugging-Face and use a threshold of 0.1 to extract the object bounding boxes.

### 4.2 HUMAN STUDY EVALUATION

The goal of our human study is to evaluate the text-guided image editing models along 3 dimensions: (i) the quality of the text-guided editing applied, (ii) the quality of other object properties preserved, and (iii) the quality of source image's overall context preserved. These dimensions mirror the 3 questions presented to human annotators, as discussed in Sec 3.2.1. In Fig 4, we visualize the scores from the top 4 editing methods for each of the three questions asked in the human study template.

**Question 1. (Editing Accuracy).** In the "Quality of Editing", which denotes the efficacy of editing, we find that Instruct-Pix2Pix, SINE and Null-Text perform the best amongst all methods. Dreambooth displays a large variation in scores across the different edit types. In particular, we also find that the human-study scores for edit types involving non-spatial changes (e.g., object-addition, object-replacement, alter-parts) are higher than edits involving spatial changes (e.g., positional-addition, size). However, we highlight that there is no one consistent 'winner' across all the edit types.

**Question 2: (Context Preservation).** For "Quality of Object Properties Preserved" and "Quality of Image Context Preserved", we find that Null-Text and Instruct-Pix2Pix fare the best across the methods. This suggests that they are better at preserving the qualitative aspects of the object and image which is an important requirement in editing. SINE and Dreambooth, on the other hand, obtain low scores on these two questions despite their high scores in editing efficacy.

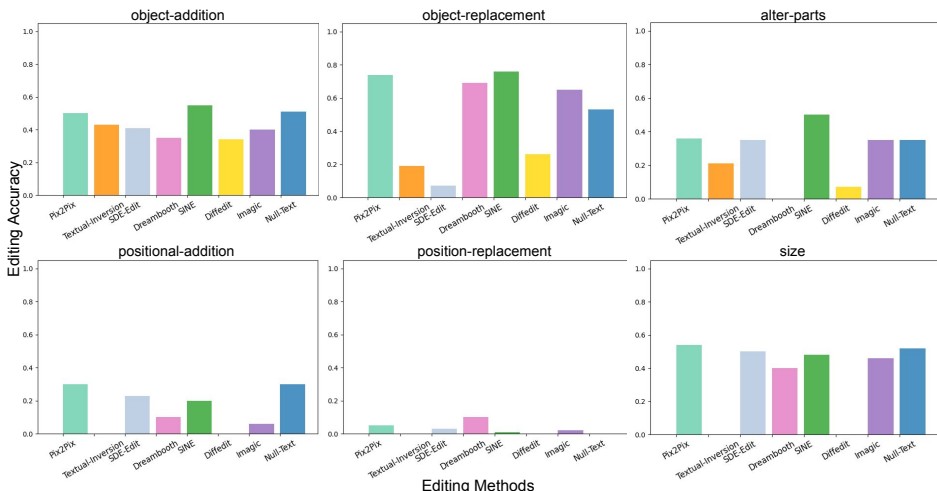

Figure 5: **Evaluation on EDITVAL using OwL-ViT across eight state-of-the-art text-guided image editing methods.** We find that while the text-guided image editing methods perform satisfactorily for edits corresponding to object manipulation, they suffer on edits requiring spatial knowledge such as `positional-addition` or `position-replacement`. Overall, we find **Instruct-Pix2Pix**, **Null-Text** and **SINE** to perform well across the majority of the editing types.

Overall, based on the human scores across these three questions, Instruct-Pix2Pix and Null-Text fare the best amongst all methods. We provide more details on the human study data collection, filtering and evaluation in Appendix G and more results in Fig 11.

### 4.3 AUTOMATED EVALUATION USING VISION-LANGUAGE MODELS

**Editing Accuracy**. We use our automated evaluation pipeline described in Sec 3.2.2 to evaluate the 8 state-of-the-art image-editing methods across 6 of the 13 edit types in $\mathcal{A}$. From our results in Fig 5, we find that the performance of most text-guided image editing methods suffer even on simple editing operations, including `object-addition` and `object-replacement`. For example, across the 8 image editing methods we evaluated, we see that their editing accuracy ranges from only 35% to 55% for `object-addition`. Of the methods, we find that SINE, Instruct-Pix2Pix and Null-Text perform the best for edit types that directly modify the object, for example `object-addition`, `object-replacement` and `alter-parts`. For `size`, on the other hand, we find Instruct-Pix2Pix performs the best, with SDE-Edit, SINE, Null-Text and Imagic also performing comparably. Although there is no clear 'winner', generally we find Instruct-Pix2Pix to be a strong text-guided image editing method for editing operations corresponding to object manipulation. We highlight that Instruct-Pix2Pix does not require any fine-tuning during the editing operation unlike other methods, including Dreambooth, SINE and Imagic[2]. For spatial editing operations such as `positional-addition` and `position-replacement`, however, we find that none of the text-guided image editing methods perform well. In particular, for `position-replacement`, we find that most of the text-guided image editing methods have a very low accuracy ranging between 0 to 15%. For `positional-addition`, the editing accuracy ranges from 0% to 30%, with Null-Text inversion and Instruct-Pix2Pix performing the best. These results show that current text-guided image editing methods are yet to handle complex editing operations which require spatial manipulation in images. We provide visual case studies corresponding to different editing methods and edit operations from EDITVAL in Appendix O and Fig 1.

**Fidelity of Edited Images to Original Images.** In Figure 7-(b), we use the DINO score (Caron et al., 2021) to assess the similarity between original and edited images across all edit types in EDITVAL. DINO scores represent the average pairwise similarity of [CLS] embeddings between these images. From these scores, we find that Textual-Inversion often leads to significant deviations from the original images. Diffedit, on the other hand, generally maintains fidelity with DINO scores exceeding

---

[2]Although Dreambooth and Textual-Inversion require more than one sample for fine-tuning, for fairness we only use one sample to be consistent across all the methods.

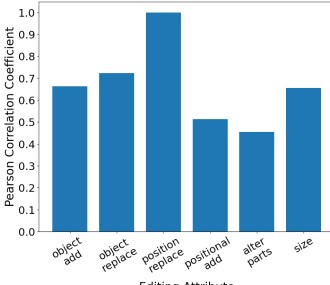 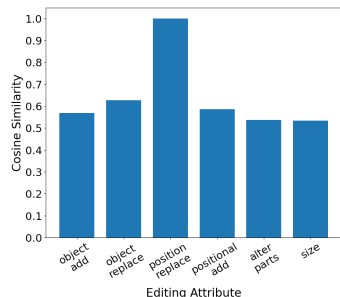

Figure 6: **EDITVAL correlation with human-score from AMT Study for six edit-types**. We obtain human annotation scores falling in the range of $\{0, 1, 2, 3\}$ for all the images involving a given edit-type; the correlation is then computed b/w these scores and EDITVAL binary scores. The general trend depicts a moderate-to-strong correlation b/w two evaluations.

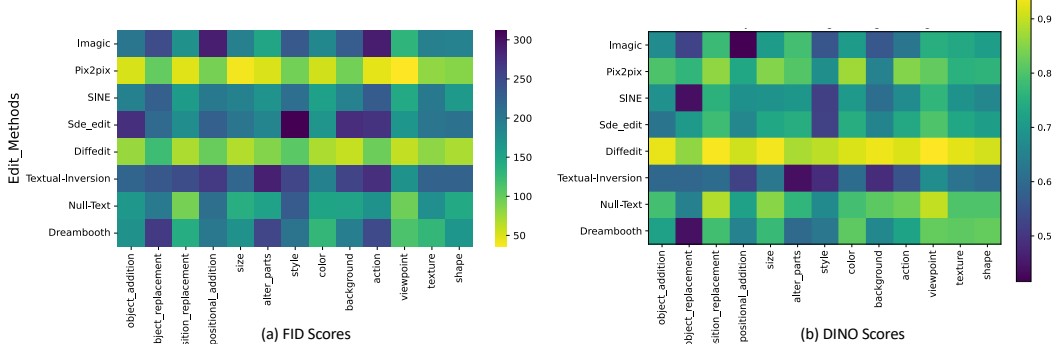

Figure 7: **Fidelity of Edited Images to Original Images.** (a) Average FID Heusel et al. (2017) score computed between the original images and the edited images across all the 8 methods tested on EDITVAL. Lower FID score quantitatively indicates a better image quality. (b) Average DINO score between the original images and the edited images across all the 8 methods tested on EDITVAL. We find that for certain methods such as Textual-Inversion, the edited images change significantly from the original images across all edit categories. For spatial changes, we find that the edited images do not change significantly across different methods.

0.85 across most categories, aligning with our human evaluation results. For complex spatial edits like `position-replacement` where methods are sensitive, edited images tend to resemble the originals. These scores show strong correlation with human evaluation in Appendix D. We also compute FID scores (Figure 7-(a)) to gauge image quality across all edit types. Instruct-Pix2Pix, followed by DiffEdit, achieves the lowest FID scores, indicating superior image quality and editing performance. Conversely, Textual-Inversion exhibits the highest FID score overall, suggesting lower image quality in the edited images. Interestingly, these results closely parallel our automated DINO score evaluation (Figure 7-(b)). Overall, we also find a strong alignment of the FID and DINO scores with the questions asked in the human study: (i) "Quality of Object Properties Preserved" and (ii) "Quality of Image Context Preserved". Diffedit and Instruct-Pix2Pix obtain low FID scores and a high DINO score signifying that the edited images do not change significantly from the original. This is similar to the human study results obtained in Fig 4–(b) and Fig 4-(c). Given robust FID computation requires a large number of samples, we recommend to use the DINO scores as a proxy for evaluating image context preservation.

> **General Takeaway.** Instruct-Pix2Pix, Null-Text and SINE are the top-performing methods on EDITVAL with both automated evaluation and human-study, with Instruct-Pix2Pix and Null-Text being better at preserving original image properties than other methods.

## 4.4 ON THE ALIGNMENT BETWEEN AUTOMATED EVALUATION AND HUMAN-STUDY

One of the primary contributions of EDITVAL is to provide an automated proxy evaluation of text-guided image editing methods for the set of edit types in $\mathcal{A}$. To validate the effectiveness of automated evaluation scores from EDITVAL, we compute their correlation with the annotation scores obtained

from our human study. In particular, we compute the correlation between human annotation score which fall within the range of $\{0, 1, 2, 3\}$ and the binary scores derived from EDITVAL for the six primary edit types. The correlation numbers are then averaged across all editing methods. We evaluate the correlation using two prominent similarity measures: (i) Pearson Coefficient Correlation and (ii) Cosine Similarity, and report results of our analysis in Fig 6. In specific, we observe that `positional-replacement` edit-type attains a perfect correlation of 1.0, indicating an accurate alignment between EDITVAL scores and human annotation scores. Other edit types also display a strong noteworthy correlations, as can be seen with `object-addition` having correlation between 0.6 and 0.7, while `positional-addition` and `alter-parts` attains only moderate correlations ranging from 0.45 to 0.6. These scores support the alignment of our automated pipeline with human ground-truth annotations.

## 5 GUIDELINES ON HOW TO USE EDITVAL

Instead of relying on a single metric such as CLIP-Score, EDITVAL strives to serve as a checklist for evaluating text-guided image editing methods. We lay down certain guidelines on how EDITVAL should be leveraged to produce a report on the efficacy of a given editing method:

- **Dataset Utilization**: Incorporate the 648 unique edit-operations encompassing multiple edit-types from our dataset along with a new image editing method.
- **Editing Accuracy Evaluation**: (i) Employ Owl-ViT rules in the automatic evaluation pipeline with our rules to obtain scores for the six edit-types. (ii) Use our standardized human-study template to acquire scores for the seven edit-types.
- **Context Preservation Evaluation:** Utilize DINO scores to gauge the efficacy of the method in preserving image context.
- **Form a Report**: Compile a comprehensive report integrating information from the earlier steps to comprehend the effectiveness of the editing method in fine-grained detail.

## 6 QUALITATIVE ANALYSIS WITH VISUAL CASE STUDIES

In our case study (detailed in Fig 1), we present qualitative examples from the evaluation of various text-guided editing methods using EDITVAL. Focused on a subset of edit types, we showcase both successful edits and instances of failure. For the `background-change` edit-type applied to an image of a "car," Instruct-Pix2Pix, Null-Text, and Dreambooth accurately replace the background, while SINE partially accomplishes the edit. SINE and Dreambooth not only change the background but also alter the original properties of the car, such as size and viewpoint, aligning with findings from our human study (Fig 4) where preserving image context after background edits is challenging. Regarding `position-replacement`, where a stop sign is moved to the center of the image, all editing methods struggle. For simpler edits like color changes, Pix2Pix performs well, but Dreambooth fails to change the boat's color in one instance, altering the background instead. SINE introduces the color but also shrinks the boat's size and changes the background. For `object-addition` or `alter-parts`, Instruct-Pix2Pix excels, applying intended edits without significant alterations. Additional case studies are covered in Appendix O. In summary, our extensive analysis of 8 text-guided image editing methods on EDITVAL reveals that while some methods excel at correct edits without altering image context, most struggle to preserve context, even with correct localized edits.

## 7 CONCLUSION

In this study, we introduce EDITVAL, a comprehensive benchmark designed to assess text-guided image editing methods using real images across diverse edit types (e.g., `object-addition`, `viewpoint`, etc). EDITVAL consists of a dataset $\mathcal{D}$, a catalog of edit types $\mathcal{A}$ with their corresponding edit operations, and evaluation procedures, offering a complete framework for evaluating text-guided image editing methods. Through rigorous evaluation, we benchmark eight state-of-the-art text-guided image editing methods, uncovering their strengths and weaknesses across various edit types. Notably, we find that no single method excels in all edit types. For instance, in object manipulation scenarios like `object-addition`, Instruct-Pix2Pix and SINE perform well, while for complex edits like `position-replacement`, most of the methods perform poorly. With its extensive range of edits and evaluation templates, EDITVAL aims to establish itself as the standard for evaluating future iterations of text-guided image editing methods.

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

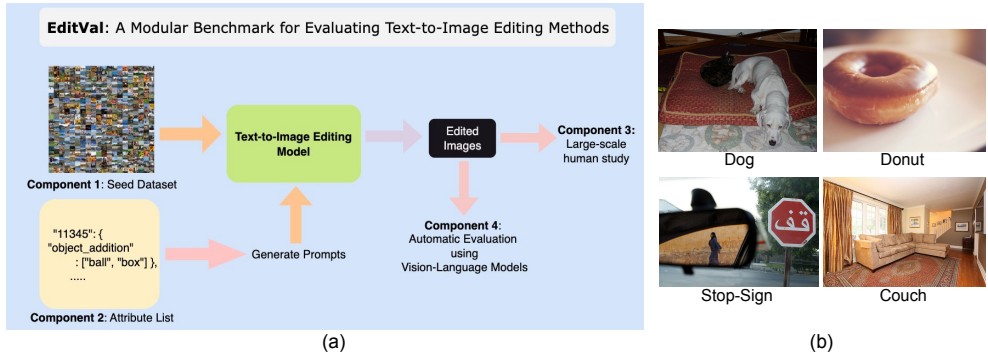

(a)                                        (b)

Figure 8: **EDITVAL pipeline consisting of a seed dataset and associated evaluation protocols.** (a) We introduce a benchmark, EDITVAL, to quantify the quality of edits in text-guided image editing methods. The benchmark consists of a dataset of images, a set of editable attributes (operations) per image, and automated and human evaluation protocols to assess the edited images. The seed dataset is curated from MS-COCO, with the set of editable attributes (operations) per image manually annotated. (b) Qualitative examples from the seed dataset selected.

## A    DESCRIPTION OF TEXT-GUIDED IMAGE EDITING METHODS

Dreambooth Ruiz et al. (2023) fine-tunes the parameters of the text-guided image diffusion model on a set of images which needs to be edited. Textual-Inversion Gal et al. (2022) fine-tunes a token embedding in the text-encoder space using a set of images. Imagic Kawar et al. (2023) edits images in three steps: (i) Fine-tunes a token embedding; (ii) Fine-tunes the parameters of the text-guided image diffusion model using the fine-tuned token embedding. (iii) Interpolation to get various edits corresponding to the target prompt. In Instruct-Pix2Pix Brooks et al. (2023), a text-to-diffusion model is pre-trained using pairs of edited images and text prompts which are generated using Prompt-to-Prompt Hertz et al. (2022). This makes Instruct-Pix2Pix training free during the editing process, hence making it fast during inference. In SDE-Edit Meng et al. (2022) – the image is corrupted using Gaussian noise which is iteratively denoised using a stochastic differential equation. In Null-text inversion Mokady et al. (2022), the unconditional text-embedding which is used for classifier-free guidance is optimized for an accurate inversion process. Using this accurate inversion process along with Prompt-to-Prompt Hertz et al. (2022) – a real image is edited. In SINE Zhang et al. (2022), a novel model-based guidance and patch-based fine-tuning process is used to edit real images. DiffEdit Couairon et al. (2022) relies on automatically locating the region of edit using the text-query by contrasting between a conditional and unconditional diffusion model.

## B    MORE DETAILS ON THE DATASET AND EDIT-TYPE SUITE

Listing 1: JSON file which contains the template of the dataset in EditVal

```
1  // editval.json template file from EditVal to create the edited
       images using text-guided-image editing methods
2  // 240698, 112345, .... : Image_Ids from MS-COCO
3  {"bench": {
4    "240698": {
5      "object-addition": [
6        {"to": ["bag", "cup", "ball", "books", "shoes"]}
7      ],
8      "positional-addition": [
9          {"to": ["bag below", "bag on top", "bag to right", "cup
                below", "cup on top", "ball to right", "ball on top",
                "books below", "books to right",...]}
10     ],
11     "texture":[
12         {"to": ["steel", "leather"]}
13     ]
```

```
14    },
15    "112345": {.....
16    }
17  }}
```

Listing 1 contains the JSON template of the dataset in EDITVAL containing 648 unique edits. To add new edit-operations or images, one simply needs to update `editval.json`.

## B.1    REASONS FOR USING MS-COCO

Our decision to use MS-COCO (Lin et al., 2014) to construct the EDITVAL benchmark is primarily motivated by the fact that it is a widely used dataset within the computer vision (and machine learning) community, including in many recent works in the text-to-image generation space (Zhang et al., 2023; Li et al., 2023). Unlike other large-scale text-image datasets like LAION, MS-COCO also provides annotations (e.g. object classes), the availability of which is critical for an automated evaluation pipeline and benchmark. The reliability of the annotations is also paramount for the robustness and reliability of the benchmark itself – in this case, MS-COCO's annotations have been validated both through a human study and from almost a decade of the dataset's usage within the research community. We note that while there are other vision datasets that would meet these criteria (e.g. ImageNet (Deng et al., 2009)), MS-COCO provides some unique advantages - namely its high image resolution (648x480 in MS-COCO compared to 469x387 for ImageNet on an average)

We also note that the images in EDITVAL are highly curated from MS-COCO, with human-in-the-loop annotators manually validating that the selected images are of a high quality and diverse for each of the 19 object classes. We show examples of the images chosen in Appendix N.

## B.2    DESCRIPTION OF EDIT TYPES

The edit types are: (i) `object-addition`: adding a new object along with an existing object; (ii) `object-replacement`: replacing a particular object; (iii) `positional addition`: adding a new object alongside an object that is already present; (iv) `size`: changing the size of an object; (v) `position-replacement`: changing the position of an object; (vi) `alter-parts`: altering a part of an existing object; (vii) `background`: changing the background of the image; (viii) `texture`: changing the texture of the image; (ix) `style`: changing the style of the image; (xi) `color`: changing the color of an object; (x) `shape`: changing shape of an object; (xii) `action`: changing action being conducted by an object; (xiii) `viewpoint`: changing the viewpoint of an existing object.

## C    IMPLEMENTATION DETAILS FOR AUTOMATIC EVALUATION

In this section, we provide additional implementation details for `object-addition`, `object-replacement`, `alter-parts`, `positional-addition`, `position-replacement`, `size`. For each of these edit-types, we compose cascaded rules which designate if an edit is correct or not. Primarily, we use OwL-ViT Minderer et al. (2022) for obtaining the object-prediction accuracies as well as their bounding boxes. The edit-type specific rules are described in Algorithm 1, Algorithm 2, Algorithm 3, Algorithm 4, Algorithm 5 and Algorithm 6. Note that `<class>` denotes the class of the original object present in the image and `<object>` denotes the class of the new object which is added.

We provide (i) working code at this anonymous link: EDITVAL code, and (ii) the details on how to run evaluation for different edit-types.

### C.1    ADDITIONAL RESULTS ON `POSITION-REPLACEMENT`

In Fig 5, we provide results corresponding to $\delta = 250$. The prompts in EDITVAL corresponding to `position-replacement` are only for carefully selected images, where the object is centered and the editing prompt requires the object to be placed to the left or right of the image. In Fig 9 – we provide further ablations on $\delta$. In particular, we choose a higher $\delta$ in our experiments, as the

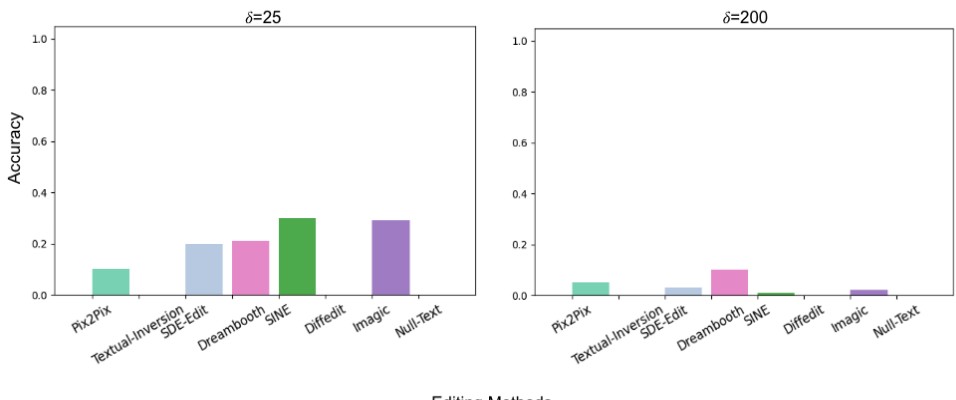

Figure 9: **For a very small** $\delta = 25$ **(Left), we find that some of the methods have an improved `position-replacement` accuracy**. However, we find that most of the editing methods do not preserve the exact position of the objects even if no prompt corresponding to `position-replacement` is given. Therefore we use a high $\delta$ value. We also provide additional results (Right) with $\delta = 200$, where we find that all the methods have very low accuracies.

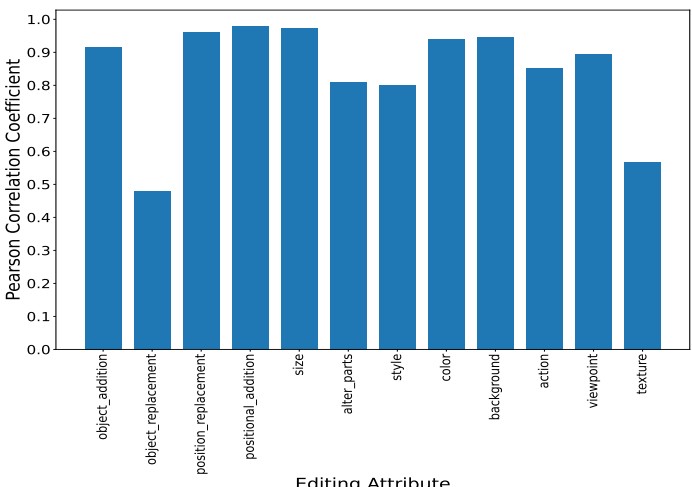

Figure 10: **Pearson Coefficient Correlation between human scores and EDITVAL DINO scores computed in automated evaluation across all edit-types.** We find that there is a strong correlation between DINO scores and human scores obtained for third parallel concerning preservation of overall image content. However, the edit-type `object-replacement` specifically has moderate correlation; overall showcasing the reliability of DINO score as a proxy metric for gauging image content preservation.

---

**Algorithm 1:** PyTorch-style pseudocode for `object-addition`

```
# Pass the edited image to OwL-ViT
c, bbox = OwL-ViT(x_edited, "a photo of <object>")
object-flag = 0
# Check if the confidence is greater than the threshold
if c > 0.1:
   # Object Found!
   print(f'Object-Added')
   object-flag = 1
else:
   print(f'Object-Not Added!')
   object-flag = 0
# End
```

---

**Algorithm 2:** PyTorch-style pseudocode for `positional-addition`

```
# Pass the edited image to OwL-ViT
c_orig, bbox_orig = OwL-ViT(x_orig, "a photo of <class>")
c, bbox = OwL-ViT(x_edited, "a photo of <object>")
object-flag = 0
# Check if the confidence is greater than the threshold
if c > 0.1 and c_orig >0.1:
   # Both objects are found!
   print(f'Both objects are found')
   if pos == 'left':
      if bbox[0] < bbox_orig[0]:
         object-flag = 1
   elif pos == 'right':
      if bbox[0] > bbox_orig[0]:
         object-flag = 1
else:
   print(f'Both object not Found !')
   object-flag = 0
# End
```

---

editing prompts explicitly requests the given objects to be placed to the left or right of the original image. Furthermore, even without `position-replacement` specific prompts, we find that the exact positions of the original objects are not preserved once passed through the diffusion model. Therefore, we recommend to use a higher $\delta$ while computing for `position-replacement`.

## D  CORRELATION BETWEEN DINO-SCORE AND HUMAN-SCORE

We previously computed DINO score between original and edited images across various edit-types to evaluate the fidelity of an edited image to the content of original image, as detailed in section 4.3. To confirm the alignment of such an *automated* evaluation to the human-study, we compute correlation between the DINO scores and the human score for third parallel, visualizing it in Fig 10. Unsurprising to us, almost all the edit-types show a strong correlation between DINO scores and human-scores, further confirming that our proposed automated evaluation is indeed a simple, reliable and quantitatively accurate way to measure the degree of preservation of original image-content in the edited image.

## E  MORE DETAILS ON EDITING TYPES AND OPERATIONS

In Table. (1), we provide a detailed description of each editing type in {*object-addition, object-replacement, action, background, shape, positional-addition, position-replacement, color, viewpoint, style, size, alter-parts, texture*} along with some examples supporting the descriptions. Overall, one

---

**Algorithm 3:**  PyTorch-style pseudocode for `object-replacement`

```
# Pass the edited image to OwL-ViT with the original class
 label
c_orig, bbox_orig = OwL-ViT(x_edited, "a photo of <class>"
# Pass the edited image to OwL-ViT with the new object
c, bbox = OwL-ViT(x_edited, "a photo of <object>")
object-flag = 0
# Check if the confidence is greater than the threshold
if c > 0.1 and c_orig > 0.1:
   # Incorrect Edit
   object-flag = 0
elif c_orig <= 0.1 and c>0.1:
   # Correct Edit - Original object absent
   print(f'New object found, old object missing!')
   object-flag = 1
# End
```

---

**Algorithm 4:**  PyTorch-style pseudocode for `alter-parts`

```
# Pass the edited image to OwL-ViT
c_orig, bbox_orig = OwL-ViT(x_edited, "a photo of <class>")
c, bbox = OwL-ViT(x_edited, "a photo of <object>")
object-flag = 0
# Check if the confidence is greater than the threshold
if c > 0.1 and c_orig > 0:
   # Checks if the alteration is within the main class object
   if bbox[0], bbox[1] inside bbox_orig:
      object-flag = 1
else:
   print(f'Object-Not Added!')
   object-flag = 0
# End
```

---

can observe that the given edit-type (e.g., *shape*) along with it's corresponding edit operation is designated in the prompt. This makes each of these edit edit-types unique in nature and ensures no overlap between them when used in a prompt. We also ensure that for each of the edit-types there are no overlaps in the edit operations. For e.g., the edit operations corresponding to *texture* (e.g., wooden, metallic) are completely disjoint from other related edit-operations involving *style* (e.g., Pointillism, Cubism) and *color* (e.g., red, yellow). For certain edit-types such as *object-addition* and *positional-addition* – there exists certain common factors such as the object which is required to be added in the scene. However, with *positional-addition*, one also mentions the position at which the given object needs to be added. This specifies the distinction between the prompts: '*Add a ball to the bench*' and '*Add a ball below the bench*' thus ensuring that no overlap exists between various editing operations.

## F   MORE DETAILS ON HUMAN STUDY EVALUATION

We also visualize the scores from the bottom 4 editing methods (as per the editing accuracy) for each of the three template questions, showcased in Fig 11. After a careful analysis of editing efficacy and the preservation of original image properties (untargeted), it is evident that even among the subpar editing methods, **Diffedit** achieves the lowest editing quality, but it manages to prevent the unintended changes related to object-properties and image-context (Fig 11(a)). On the flip side, **Imagic** despite having slightly improved editing quality struggle to effectively prevent such unintended changes.

As a part of further investigation, we exhaustively visualize the distribution of human evaluation scores (score levels ranging from 0 to 3) for all editing methods and remaining 8 edit-types from set $\mathcal{A}$ in EDITVAL in the Fig 12 and Fig 13. Under the first parallel of Fig 12, it is visually clear that the

---

**Algorithm 5:** PyTorch-style pseudocode for `position-replacement`

---

```
# Pass the edited image to OwL-ViT
c_orig, bbox_orig = OwL-ViT(x_orig, "a photo of <class>")
c, bbox = OwL-ViT(x_edited, "a photo of <class>")
object-flag = 0
# Check if the confidence is greater than the threshold
if c > 0.1:
   # Object Found!
   print(f'Object-Added')
   if pos == 'left':
      # Check with position; δ is a hyper-parameter
      if bbox[0] < (bbox_orig - δ):
         object-flag = 1
   elif pos == 'right':
      # Check with position; δ is a hyper-parameter
      if bbox[0] > (bbox_orig + δ):
         object-flag = 1
else:
   print(f'Object-Not Added!')
   object-flag = 0
# End
```

---

**Algorithm 6:** PyTorch-style pseudocode for `size`

---

```
# Pass the edited image to OwL-ViT
c_orig, bbox_orig = OwL-ViT(x_orig, "a photo of <class>")
c, bbox = OwL-ViT(x_edited, "a photo of <class>")
object-flag = 0
# Check if the confidence is greater than the threshold
if c > 0.1:
   # Compute the area of the bounding box
   area-orig = compute-area(bbox_orig)
   area-object = compute-area(bbox)
   if size == 'small':
      if area-object < (area-orig -δ):
         object-flag = 1
   if size == 'large':
      if area-object > (area-orig + δ):
         object-flag = 1
else:
   print(f'Object-Not Added!')
   object-flag = 0
# End
```

---

edit-types related to object-properties such as `color`, `texture` and so on, follow a similar trend in annotations; For these edit-types, most of the editing methods are able to moderately edit the image showing fidelity to text-instruction. Among all the editing-methods, Instruct-Pix2Pix (light-green) and Null-Text Inversion (dark-blue) are consistently able to achieve moderate to perfect editing on these object-centric edit-types. However, for the other two parallels, mostly all edit-types follow identical trend in preserving object properties and image content. Using AMT human-annotations, we further try to understand the image editing fidelity and preservation of image-content for the complex edit operations, such as changing viewpoint or action of the object-of-interest, altering the background except the primary object and so on. We here observed that changing `action` and `background` are particularly difficult and ambiguous for all editing methods. Apart from these two, as we look closer at `viewpoint` and `position-replacement`, we can state that SINE, Dreambooth and Textual Inversion are able to achieve moderate editing as per human-judgement, but they also suffer on reliability scale to preserve untargeted changes (Column:2 and Column:3 of

| Description of Editing Operations | | | | |
|---|---|---|---|---|
| Edit Type | Description | Example 1 | Example 2 | Example 3 |
| Object-Addition | Adding a new object to a scene | add *bag* to bench | add *tray* along with cup | add *tissue roll* to sink |
| Positional-Addition | Adding a new object at a particular position in a scene | *bag* below a bench | *balls* on top of bench | *bag* to right of person |
| Positional-Replacement | Replacing the position of an existing object in a scene | *donut* to left | *cat* to the right | *dog* to the left |
| Texture | Replacing the texture of an object | *wooden* bicycle | *metallic* chair | *zebra stripes* apple |
| Shape | Replacing the shape of an existing object | *duffel* bag | *hydration* backpack | *square pizza* |
| Size | Changing the size of an object in a scene | *small* pizza | *large* pizza | *small* cup |
| Style | Changing the style of a scene | cat in *realism* style | cat in *fauvism* style | potten plant in *Pointillism* |
| Alter-Parts | Altering parts of an object | add *chocolate toppings* to donut | add *tomato* toppings to pizza | add *jelly beans* to pizza |
| Object-Replacement | Replace an existing object with a new object | replace chair with *bench* | replace car with *motorcycle* | replace car with *pickup truck* |
| Viewpoint | Change the viewpoint of an object | viewpoint of chair to *front* | viewpoint of chair to *back* | viewpoint of airplane to *rear* |
| Color | Changing the color of an object | *red* backpack | *blue* backpack | *red* boat |
| Background | Changing the background of a scene | change background to *house* | change background to *forest* | change background to *street* |
| Action | Changing the action of an animal | person *standing* | person *sitting* | person *jump* |

Table 1: **Description of the different edit-types in EDITVAL.**

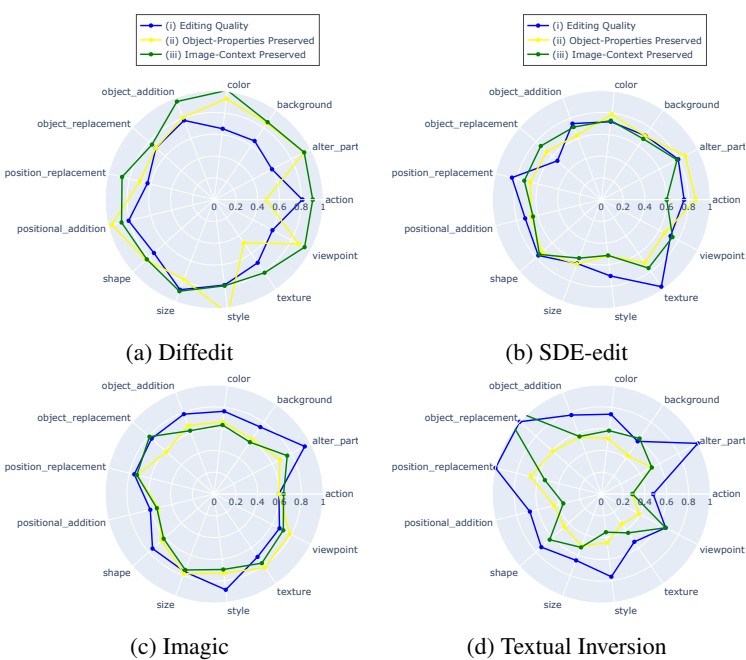

(a) Diffedit  (b) SDE-edit  (c) Imagic  (d) Textual Inversion

Figure 11: **Human Study Results for bottom 4 methods (with respect to editing accuracy) across *three* questions in the human study template.** (i) *Editing Quality*: We find that **Imagic** on average performs better editing among the bottom 4 methods (ii) *Object-Properties Preserved:* **Diffedit** is a clear winner in terms of preserving object properties; (iii) *Image-Context Preserved:* **Diffedit** again fares well in preserving the context of the original images.

Fig 13). Hence, this kind of ambiguity in varying performance on the complex edit-types makes it challenging to pick a single winning editing method.

## G QUALITY CHECK ON HUMAN ANNOTATIONS

**Eliminate Malicious Workers**: In our AMT study, we only select workers who have a HIT approval rate of greater than 90 and their number of HIT approvals is $> 500$ in the past. Each task is active for 7 days for sufficient visibility and after accepting the task, a worker is allowed ample time of 30 minutes to make their selections; it should be noted that we pay $0.3 for each completed task. As mentioned earlier, we assign a task to three unique workers and additionally, approve and pay the workers only after verifying their annotation quality. All these measures are taken into account for better quality control over human annotations. It took us 5-7 days to obtain AMT final annotations and incurred an overall cost of $1350.

**Gold Set for Verification of Annotation Quality**: Given that our AMT study tasks often require some minimum attention and effort to answer correctly, it seems logical to filter out workers who provide low-quality or almost random responses. To accomplish this, we manually respond to a total of 150 tasks within the study, forming what we refer to as our "gold set". By comparing the

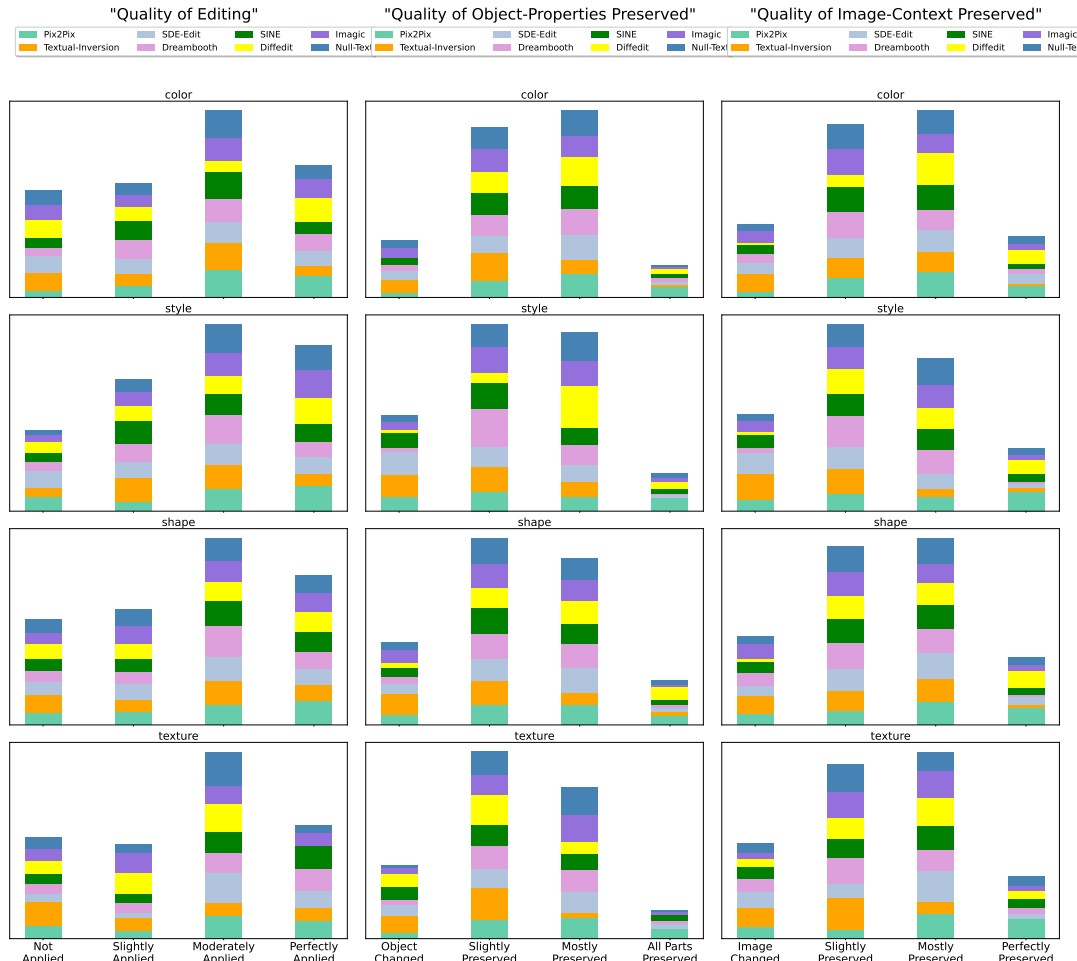

Figure 12: **Human Study Evaluation for `color`, `size`, `shape` and `texture`**: Distribution of %age of edited images falling under four levels of human annotations.

answers provided by workers to these tasks against our own responses, we eliminate those whose answers do not sufficiently align. To ensure that we do not unjustly remove valuable workers based on a single instance of poor performance, we only exclude those who exhibit subpar responses in at least three tasks from the gold set.

**Average User Agreement**: As we know that assessing the image-editing fidelty is subjective, and the annotations can sometimes vary largely among the human-subjects. Therefore, we define the *user-agreement* between workers as the percentage of workers that agree upon a single annotation-level or score for a given question. We observed in the human study that, for all the editing methods, atleast 76-78% of the the task-assignments (HIT) have a majority consensus on an answer across all three workers assigned to a task. This percentage of agreement is also consistent across all the 13 edit-types in our benchmark.

## H  VALIDATION OF CHATGPT PROMPTS

To extract the seed dataset $\mathcal{D}$ from MS-COCO corresponding to the defined editing types, we use ChatGPT in:

(i) Extracting the list of classes $\mathcal{C}$ in which the given editing operation $a_i$ is practical. For e.g., for object-addition, we prompt ChatGPT with : *'List the classes in MS-COCO on which object-addition*

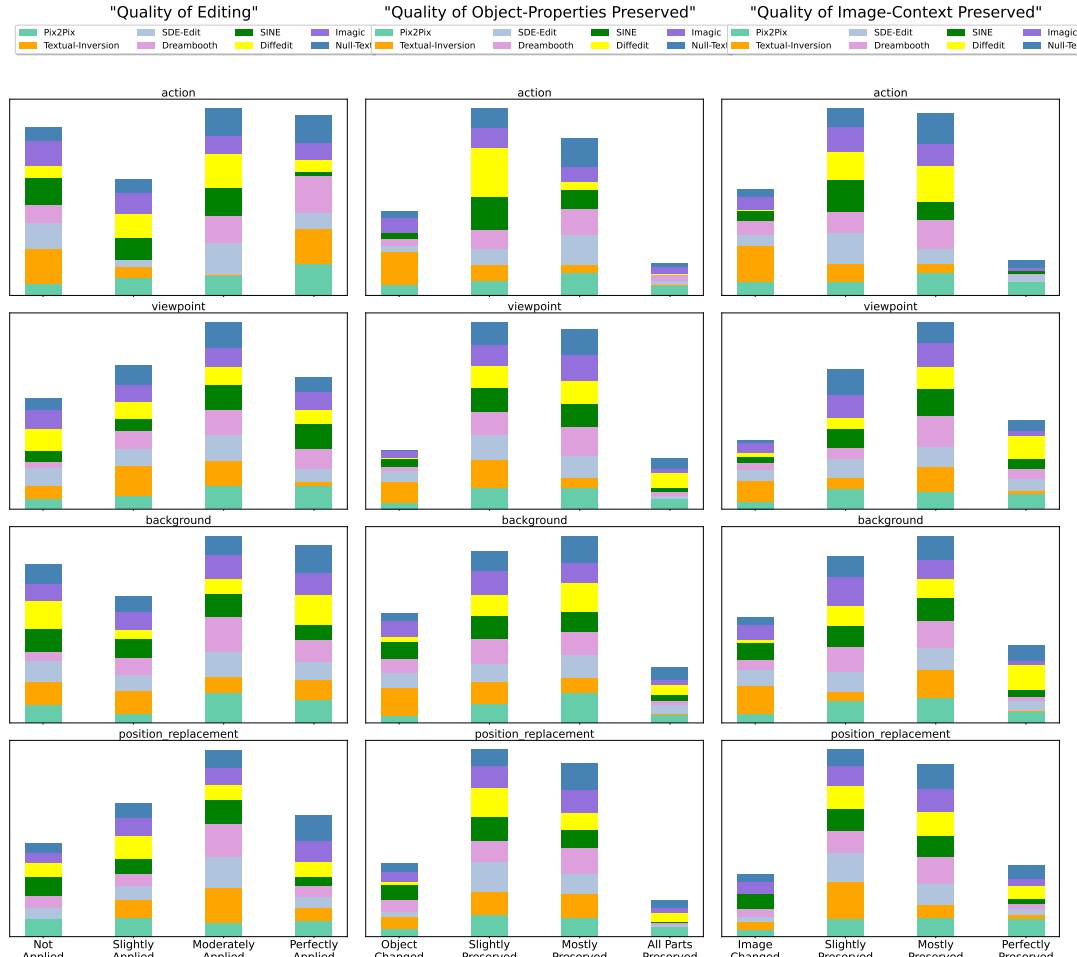

Figure 13: **Human Study Evaluation for `action`, `viewpoint`, `background` and `position-replacement`**: Distribution of %age of edited images falling under four levels of human annotations.

*is plausible'*. To validate that these classes are indeed practical to apply the edit operation on, we ask a human rater from our team to validate the results. For e.g., for object-addition, ChatGPT selects {*bench, pizza, cup, sink, person*} which are valid classes in which a new object can be added. In Table.(2), we show that human-raters agree with ChatGPT's results with 100% efficacy. We believe that the answers from ChatGPT align strongly with humans, as the prompts are simple in nature.

(ii) Once these classes are extracted, we filter 19 classes amongst them to maximize the overlap amongst different editing types. Next, for each edit dimension $a_i$ and class $c \in \mathcal{C}$, we prompt ChatGPT with a curated prompt to define the edit-operation. For e.g., in the case of object-addition for a bench, we ask ChatGPT: *What are some of the objects which can be added to a bench?* . Human-raters from our team, then manually select a subset of the answers (which are realistic) to define the edit-operation for the particular class $c$. Therefore, in this case, we don't perform a human-study but instead use a human-in-the-loop to design the edit operation for each class $c \in \mathcal{C}$ and edit-type $a_i$.

Given that we use a small human study to validate the classes used in our benchmark and also use a human-rater in the loop for defining the edit operation, our benchmark design is robust and does not consist on unrealistic editing operations.

| Edit-type | Human Study Score | | | |
|---|---|---|---|---|
| | Human Rater 1 | Human Rater 2 | Human Rater 3 | Human Rater 4 |
| Object-Addition | 100% | 100% | 100% | 100% |
| Positional-Addition | 100% | 90% | 100% | 95% |
| Positional-Replacement | 100% | 95% | 100% | 100% |
| Texture | 100% | 100% | 90% | 100% |
| Shape | 100% | 100% | 100% | 100% |
| Size | 100% | 95% | 100% | 100% |
| Style | 100% | 90% | 100% | 100% |
| Alter-Parts | 100% | 100% | 100% | 100% |
| Object-Replacement | 100% | 100% | 95% | 100% |
| Viewpoint | 100% | 95% | 100% | 95% |
| Color | 100% | 100% | 100% | 100% |
| Background | 100% | 100% | 100% | 100% |
| Action | 100% | 100% | 100% | 100% |

Table 2: **Agreement of Human Raters with ChatGPT prompts.**

| Method | Description of Prompt Design | | | |
|---|---|---|---|---|
| | Object-Addition | Color | Positional-Addition | Viewpoint |
| Instruct-Pix2Pix | Add a ball to the bench | Change the color of the bench to brown | Add a ball to the left of bench | bench from the back viewpoint |
| SDE-Edit | Add a ball to the bench | Change the color of the bench to brown | Add a ball to the left of bench | bench from the back viewpoint |
| Textual-Inversion | A ball along with [V*] bench | A brown [V*] bench | A ball to the left of [V*] bench | [V*] bench from the back viewpoint |
| Dreambooth | A ball along with [V*] bench | A brown [V*] bench | A ball to the left of [V*] bench | [V*] bench from the back viewpoint |
| SINE | A ball along with bench | A brown bench | A ball to the left of bench | bench from the back viewpoint |
| Diff-Edit | A ball along with the bench | A brown bench | A ball to the left of bench | bench from the back viewpoint |
| Null-Text | A ball along with the bench | A brown bench | A ball to the left of bench | bench from the back viewpoint |
| Imagic | A ball along with the bench | A brown bench | A ball to the left of bench | bench from the back viewpoint |

Table 3: **Examples of Prompt Design for the Different Editing Methods on EDITVAL.**

# I   CONSISTENCY AMONGST PROMPTS FOR GENERATING EDITED IMAGES

In Table. (3), we show qualitative examples of various prompts which are used for generating the edited images. We highlight that different state-of-the-art text-guided editing methods require different style of prompt curation. For e.g., for Instruct-Pix2Pix, the prompt is in the form of instruction, whereas for methods such as SINE or Imagic, the prompt is non-instruction based. While it is infeasible to define exact instructions or prompts for different editing methods due to their inherent technical design, our benchmark EDITVAL standardizes the edit-type (e.g., object-addition) and the specific editing operation (e.g., adding a ball) around which an editing prompt can be defined for distinct methods. This enables design of prompts or instructions which are similar to one another. For e.g., in Table. (3), we show that there exists high similarities between the instructions and prompts used across the different editing methods tested in our benchmark.

# J   ISSUES WITH CLIP FOR EVALUATING SPATIAL EDITS

For evaluating edited images on our benchmark, we use OwL-ViT instead of CLIP-Score. Given that EDITVAL consists of edit-types encompassing spatial edit-types such as positional-addition or position-replacement, we use a vision-language model which has high fidelity to detecting spatial changes. To test if CLIP can correctly evaluate spatial edits, we simulated the editing use-case of positional-addition, where a new object is added to an already existing object. From MS-COCO, we extract a set of images $\mathcal{I}$ (size of 100) which has annotations about at least 2 objects $o_1^i$ and $o_2^i$ for a given image $x_i \in \mathcal{I}$. For each image $x_i \in \mathcal{I}$, we create two captions: (i) $c_1^i = o_1^i$ *to the left of* $o_2^i$; (ii) $c_2^i = o_1^i$ *to the right of* $o_2^i$. The objective is to classify the image $x_i$ to the correct caption between $c_1^i$ and $c_2^i$. From Table. (4), we find that CLIP lags behind OwL-ViT for evaluating such spatial edit-types. For ground-truth captions, where an object is to the Left or an object is to the Right of another, CLIP fails to detect this. However, OwL-ViT has a good performance indicating it is a good choice for evaluating spatial edit-types.

| Method | Left | Right |
|--------|------|-------|
| CLIP | 55.4 | 56.8 |
| OwL-ViT | **87.1** | **88.5** |

Table 4: **CLIP vs. OwL-ViT for evaluating spatial edit-types.** We evaluate CLIP and OwL-ViT for spatial edit-types on a small subset of MS-COCO, where the ground-truth has : (i) one object to the Left of another; or (ii) one object to the Right of another.

## J.1  NOTE ON CLIP-SCORE CORRELATION WITH HUMAN STUDY

For the spatial edits involving `position-replacement` and `positional-addition`, we find the correlation of CLIP-Score with the human-study scores.  Overall for `position-replacement` we find a correlation of 0.24 and for `positional-addition`, we find a correlation of 0.19 (averaged across all 8 editing methods). These correlations are lower than OwL-ViT, therefore we choose OwL-ViT as our evaluation metric.

## K  NOTE ON HUMAN-STUDY CORRELATION WITH OTHER EDIT-TYPES

For edit-types involving *viewpoint, color, action, texture, shape, style*, we compare the scores from OwL-ViT with the human-study results, but we observe a poor correlations of 0.15, 0.21, 0.29, 0.22, 0.34 respectively. This is averaged across a subset of methods: *Instruct-Pix2Pix, SINE and Imagic*. Given this poor correlation with the human-study, we suggest to only use automated evaluation for the 6 edit-types as highlighted in the paper. The primary reason for poor correlations is that current vision-language models are not robust towards understanding viewpoint, action or shape effectively. For CLIP, we find similar low correlation scores for these edit-types. Given that OwL-ViT is pre-trained with a CLIP-like objective on similar data-scales, these are expected results. The main advantage of OwL-ViT is in it's ability to provide bounding boxes for open-vocabulary classes, thereby enabling us to evaluate on spatial edit-types more effectively, as bounding boxes provide additional signals.

## L  COMPARISON WITH EDITBENCH

While related to our work, our benchmark EditVal extends the work of EditBench (Wang et al., 2023) in 4 key ways: First, EditBench can only be used to evaluate text-guided image in-painting methods and requires a mask to be input along with the image to be edited and the text prompt. In comparison, EditVal requires just the image and text prompt to be provided, and can therefore be flexibly used to evaluate any text-guided image editing method. Second, EditBench only supports non-spatial edit operations for object and scene manipulations. In comparison, EditVal spans 13 unique edit types encompassing both spatial and non-spatial edits thus providing a more comprehensive and fine-grained understanding of the successes and the failures of the current generation of text-guided image editing methods. Third, EditBench relies only on a human study to provide a score for each edit. In comparison, EditVal leverages both automated evaluation and a human study, with our results showing that our automated evaluation is highly correlated with scores provided by human annotators. We also highlight that unlike the human evaluation protocol in EditBench, our human study protocol has been standardized in such a way that it can be easily extended to any text-guided image-editing method. Compared to EditBench, our empirical study evaluates a wider and more diverse set of 8 SoTA text-guided editing methods (e.g., Pix2Pix, SINE). With all these advantages, we believe that EditVal can more flexibly be adopted by the research community and can provide much finer-grained insights into the generative abilities of image-editing methods.

## M  STANDARD EVALUATION OF IMAGE QUALITY

As a standard benchmark, we compute FID score Heusel et al. (2017) to access the image quality using the set of original and edited images across all the edit types in EDITVAL. FID score precisely computes the fidelty of edited image in the latent space of a generative model w.r.t to the distribution

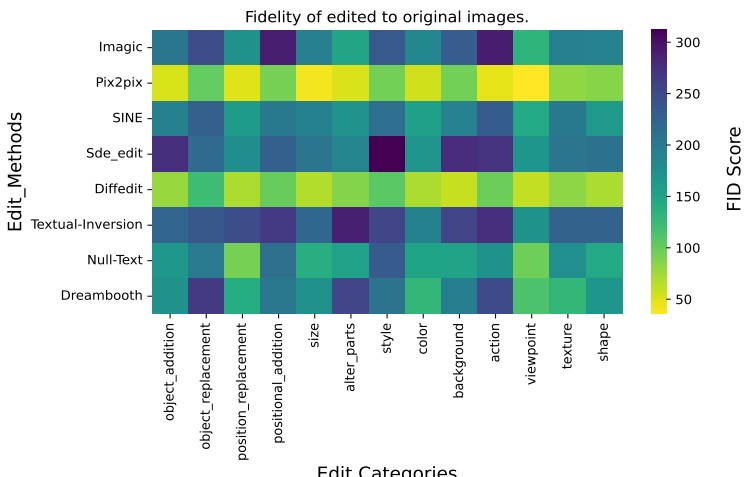

Figure 14: **FID Heusel et al. (2017) score computed between the original images and the edited images across all the 8 methods tested on EDITVAL**. Lower FID score quantitatively indicates a better image quality. We calculate FID score by computing Frechet distance between two Gaussians fitted on the set of original and edited images. By default, Heusel et al. (2017) works with the final layer features from Inception network, but, due to limited samples, we can use a shallow feature-layer even though the FID score may no longer reflect the visual quality accurately.

of a set of real images. As we can clearly observe in the Fig 14, Instruct-Pix2Pix (closely followed by DiffEdit) achieves the best (lowest) FID score and performs the best editing in terms of image quality; whereas Textual-Inversion has highest FID score overall, indicating that the edited images are of inferior quality. Interestingly, these results show close resemblance to what we observed earlier in our automated evaluation of image-fidelity by computing DINO scores shown in Fig 7.

# N    VISUALIZATION OF IMAGES IN EDITVAL

In Fig 15 and Fig 16, we show qualitative examples of the images in EDITVAL. Our human-in-the-loop process ensures that the images selected in each class are diverse and distinct in characteristics, therefore providing a comprehensive test-bed for evaluating text-guided image editing methods.

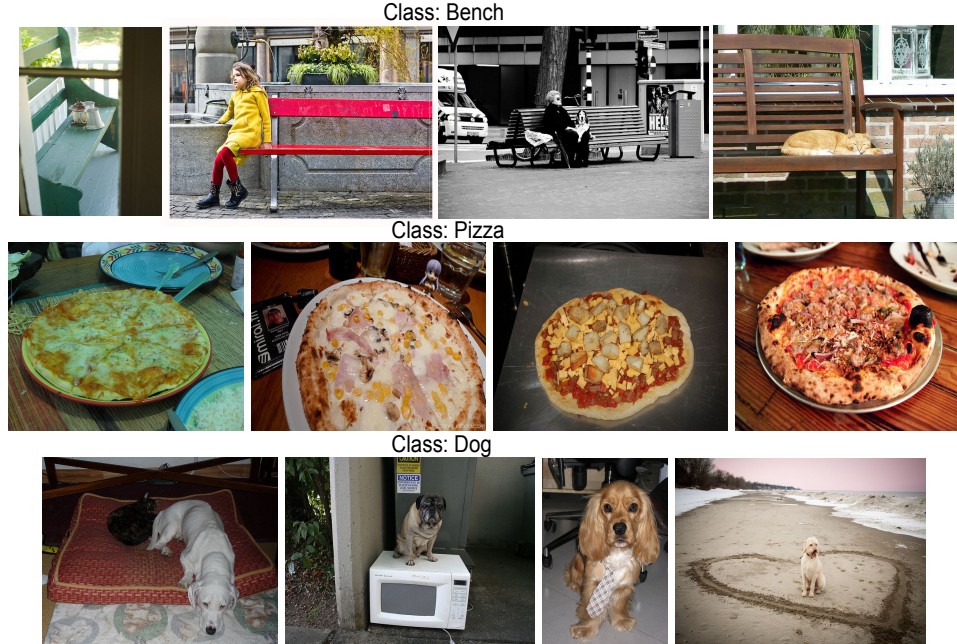

Figure 15: **Images in EditVal**: Representative Images from the classes : *Bench, Pizza, Dog*. We ensure that the images selected in each class are diverse. For e.g., for the *Bench* class, the color or the type of the bench is distinct. Similarly for the *dog* class, all the dogs are of different breeds. For the *pizza* class, we ensure that the toppings are distinct.

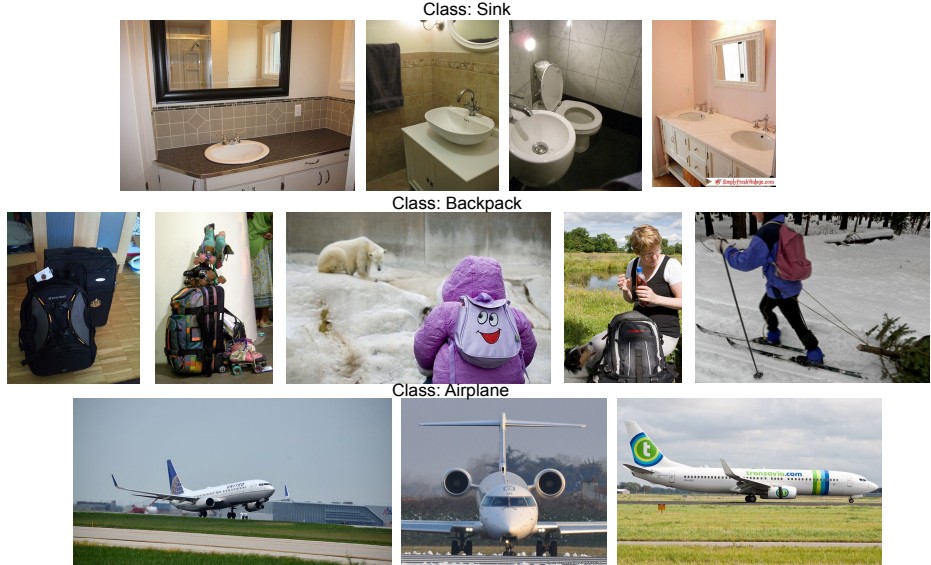

Figure 16: **Images in EditVal**: Representative Images from the classes : *Sink, Backpack, Airplane*. For all the classes, we ensure that the images are diverse. For the *sink* class, the images have sinks in different viewpoints. For *backpack* class, all the bags are of different types / color. For the *airplane* class, we ensure that the viewpoints are different. Our human-in-the-loop process while creating the dataset ensures diversity amongst the images.

## O VISUAL CASE STUDIES

In Fig 17, Fig 18, Fig 19, Fig 20, we provide different visual cases-studies corresponding to a subset of edit operations across different text-guided image-editing methods.

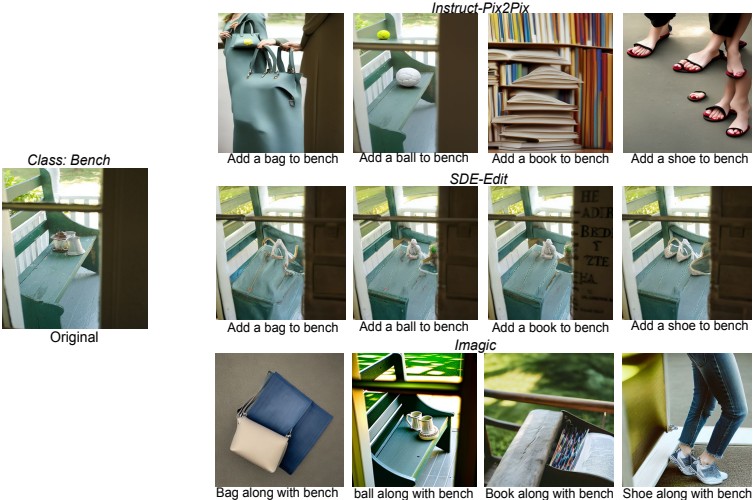

Figure 17: **Visual Case Studies : Object-Addition**: For the *bench* image in the case of Pix2Pix and Imagic, we can observe that whenever the new object gets added correctly, the edited image often omits the *bench* object across all the methods, highlighting that existing methods suffer on simple edit operations such as *object-addition*. For SDE-Edit, we find that the correct object does not get added to the edited image.

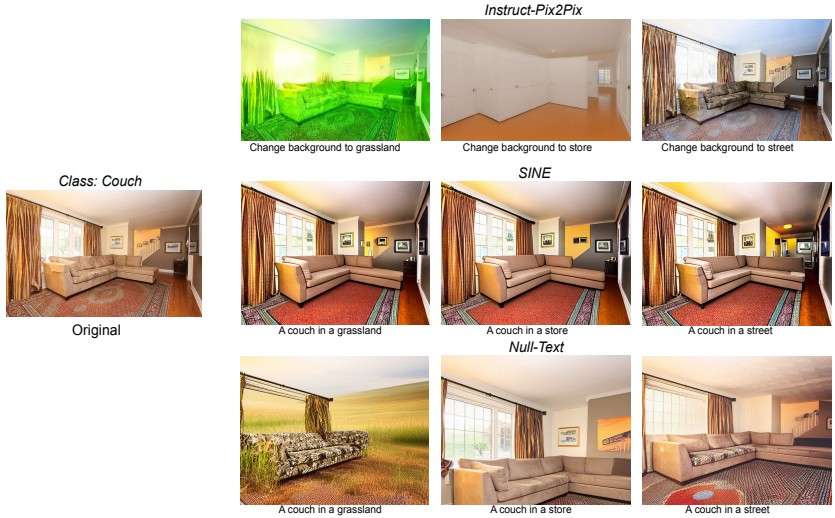

Figure 18: **Visual Case Studies : Background**: For the *couch* image, we can observe that all the methods fail at inserting the correct background behind the couch. Instruct-Pix2Pix inserts a shade of a grassland behind the couch, whereas Null-Text is able to correctly place the couch in a grassland, though the shape and characteristics of the couch change drastically. For SINE, we find that none of the background-edit operations are correct and the edited images are closed to the original image.

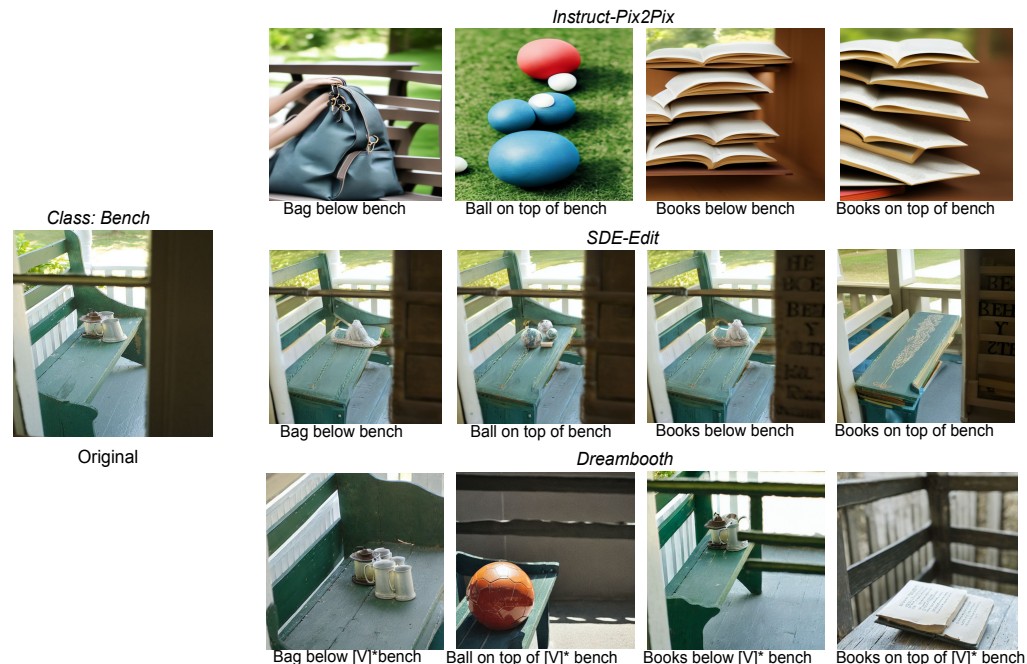

Figure 19: **Visual Case Studies : Positional-Addition**: For the *bench* image, we find that for InstructPix2Pix – the correct object is added, but the spatial positioning is not respected. For SDE-Edit, only for the case of 'ball' on top of the bench, the edit is correct. For the other cases, the correct object is also not added. This is similar to what we observed for object-addition in the case of SDE-Edit. For Dreambooth, we find that the correct spatial positioning is respected in two cases, but the structure of the original bench changes drastically.

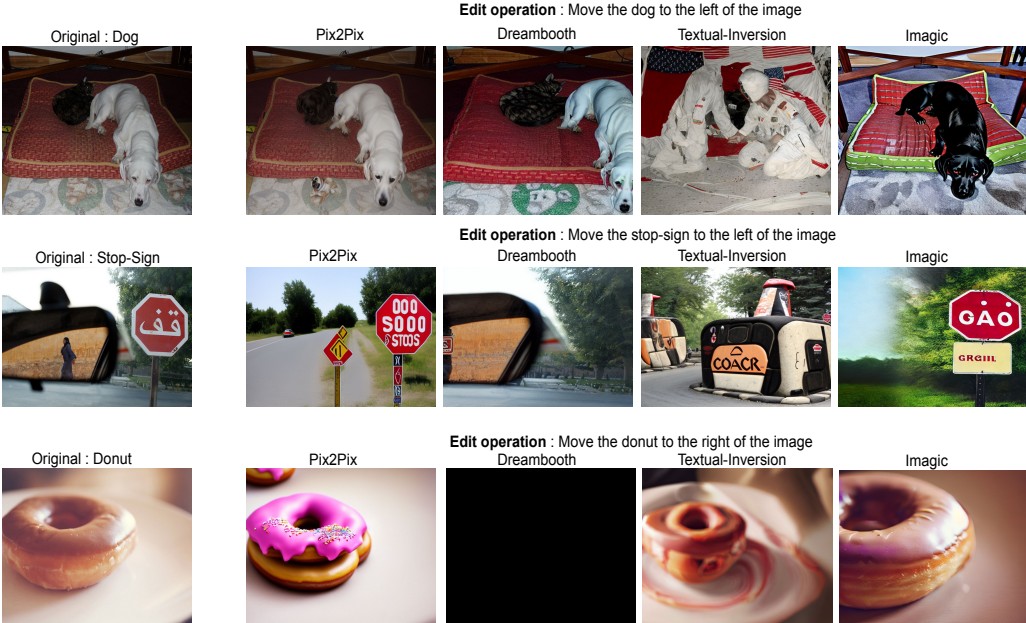

Figure 20: **Visual Case Studies : Position-Replacement**: For each of the methods including Pix2Pix, Dreambooth, Textual-Inversion and Imagic, we find that the post-edited images don't respect the spatial edit instruction given in the prompts. For textual-inversion, we find that the final edited images change drastically when compared to the original images. For Pix2Pix, the output edited images contain the old objects, but no spatial changes take place. For Imagic, spatial edit fidelity is not followed. In all, we find that current text-guided image-editing methods struggle on edit operations involving spatial changes. In the case of Dreambooth, we find that except for the case of Donut, where the image is black due to NSFW filter, the edited images from dog and stop-sign classes don't follow spatial-edit fidelity.

