# OpenReview forum: "Benchmarking Diffusion Based Text-Guided Image Editing Methods"
_ICLR.cc/2024/Conference — Submitted to ICLR 2024_

### Official Review · Reviewer_b2qM · 2023-10-28

**Soundness:** 3 good
**Presentation:** 3 good
**Contribution:** 2 fair
**Rating:** 6
**Confidence:** 3

**Summary:**

EditVal proposes a standardized benchmark for evaluating text2image editing methods across various edit types. The proposed benchmark has an automated evaluation pipeline and enables evaluation in scale. The paper benchmarks 8 SoTA editing methods using the proposed benchmark and finds that the benchmark positively correlates to human eval. This study found there is no clear winner in all categories and it discovered that all SoTA methods perform poorly for complex editing operations.

**Strengths:**

– the paper is well-written and well-motivated, trying to standardize evaluation on text-based image editing methods.
– the proposed benchmark is general purpose and includes larger and more complete edit types compared to previous benchmarks.
– the proposed method adopts OwL-ViT for evaluating edit types that require fine-drained localization capability

**Weaknesses:**

– the proposed benchmark only includes real images from MSCOCO, a missing evaluation on common use case is editing on synthetic generated images, which I think should be added to the benchmark
– the automatic evaluation pipeline cannot capture hallucinations, i.e. Figure 1 object addition, Dreambooth added a plausible wine glass next to the pizza, but the original content does not preserve well, and Pix2Pix is the opposite;

**Questions:**

– what are your thoughts on the practical usage of the benchmark? does achieving a high score translate into a better editing method in practice or does it just mean it is relatively better than other methods?
- I do have concerns on the practical usefulness of the benchmark, especially the automatic evaluation pipeline, hopefully the authors can address them as mention in the weakness

---

> ### Author Response · Authors · 2023-11-20
> **Response to Reviewer**
>
> We thank the reviewer for the constructive comments, especially that the problem is important and well-motivated given a lack of standardized editing benchmarks!
>
> Below we provide responses to the reviewer's questions:
>
> **the proposed benchmark only includes real images from MSCOCO, a missing evaluation on common use case is editing on synthetic generated images, which I think should be added to the benchmark – the automatic evaluation pipeline cannot capture hallucinations, i.e. Figure 1 object addition,....** : We thank the reviewer for these ideas - they are interesting! The scope of our paper was to develop an evaluation pipeline to evaluate edits on real images. Although our pipeline could plausibly be extended to evaluating edits in synthetically generated images, we did not explicitly consider these in our work. We also note that expanding the size of the evaluation dataset can significantly increase the evaluation time, so our choice of dataset size was carefully considered.  For this reason, we consider an extension to synthetically generated images as future work.
>
> For the reviewer’s second point on understanding if the original content is preserved or not, we highlight that the automatic evaluation does contain computing the DINO scores in conjunction with the scores from OwL-ViT. These DINO scores are proxies for the third question in human-study which captures if the original image content is preserved well or not. We have updated Sec. (3.1) to better reflect this. Indeed, we find that it displays a strong correlation with human-study as well (see Sec. 4.3 and Sec. (D)). We have made these points more clear in the main paper and provided additional guidelines in the form of a checklist (see updated Sec. (5)) on how our framework can be used to evaluate text-guided editing methods.
>
> **what are your thoughts on the practical usage of the benchmark? does achieving a high score translate into a better editing method in practice or does it just mean it is relatively better than other methods? ; I do have concerns on the practical usefulness of the benchmark..** : Current text-guided image editing methods predominantly hinge on a singular metric, the CLIP-Score. Additionally, there is no single standardized dataset that covers a diverse array of edit operations (e.g. object-addition or changing the location of an object) which further compounds the evaluation challenge. Existing works often resort to human studies, yet the templates employed vary across papers, making fair comparison difficult.
> Our paper addresses these challenges by presenting a comprehensive standardized package, inclusive of a dataset and two evaluation schemes, designed to capture different facets of text-guided image editing methods. Our benchmark's evaluations offer richer insights than CLIP-Scores, probing methods across various dimensions of edits. Moreover, our human-study template can seamlessly adapt to future editing methods without the reliance on edited images from preceding methods.
> Our benchmark can be employed as a practical checklist or guide for evaluating text-guided editing methods. We expound on these points in the main paper (refer to Sec. 5) to clarify how EditVal can be practically utilized. In essence, we propose the following use of EditVal as a checklist:
> (i) Dataset Utilization: Incorporate the 648 edit-operations from our dataset along with a new image editing method.
> (ii) Editing Accuracy Evaluation:
> (a) Employ Owl-ViT rules in the automatic evaluation pipeline to obtain scores for the six edit-types.
> (b)Use our standardized human-study template to acquire scores for the seven edit-types.
> (iii) Context Preservation Assessment: Utilize DINO and FID to gauge the efficacy of the method in preserving image context.
> (iv) Compile a comprehensive report integrating information from Steps 1, 2, and 3 to comprehend the effectiveness of the editing method in fine-grained detail.
>
> We contend that relying on a single score is inadequate, as exemplified by Fig. 5, where SDE-Edit outperforms Dreambooth in object-addition but lags behind in object-replacement. A singular score fails to unveil such nuanced details about a method, underscoring the importance of our proposed comprehensive benchmark report for a more thorough understanding of editing method efficacy.
> We want to highlight another important point in terms of practical usage : We have open-sourced our code at https://github.com/deep-ml-research/editval_code, will maintain a leaderboard for different edit-types at https://deep-ml-research.github.io/editval/ and our dataset is extremely easy to use with a COCO-dataloader used in conjunction with a json file (as shown in Sec. (B)).

---

> > ### Author Response · Authors · 2023-11-22
> > **Looking Forward To Your Response!**
> >
> > We thank you again for your valuable feedback and comments which has helped to strengthen our paper. As the discussion period is ending soon, we would really appreciate if you could let us know if our responses have addressed your concerns. We will be happy to answer any further questions and address any remaining concerns to the best of our abilities in the remaining time!

---

### Official Review · Reviewer_mpAW · 2023-10-30

**Soundness:** 3 good
**Presentation:** 3 good
**Contribution:** 2 fair
**Rating:** 5
**Confidence:** 4

**Summary:**

The paper introduces a standardised benchmark for text-driven image-editing methods. The authors evaluate eight s.o.t.a. editing methods and analyse their performance across popular editing tasks.

**Strengths:**

* The benchmark provides a comprehensive list of edits and classes.
* Extensive human study for s.o.t.a. text-guided image-editing methods.
* Valuable conclusions about existing editing methods.
* The paper provides many thorough details and discussions explaining the design choices.

**Weaknesses:**

The automated evaluation pipeline seems incomplete and limited:
* Only 6 out of 13 edits are supported.
* The pipeline can only check the object presence, location and size. For example, it cannot recognize if a cup stands naturally on the bench for “add a cup on the bench”. These aspects make this evaluator quite vulnerable.
* It does not evaluate image fidelity and image-context preservation. One still needs FID and DINO/CLIP scores which are not specifically designed for editing evaluation.

Therefore, I do not fully understand the value of the proposed automated pipeline if one still needs a human study for reliable evaluation.
Probably, it would be reasonable to finetune some visual language model on the collected human scores and obtain something similar to ImageReward[1] but for the editing tasks. In addition, one can consider combining it with the proposed detector-based algorithms.

[1] Xu et al., ImageReward: Learning and Evaluating Human Preferences for Text-to-Image Generation

**Questions:**

* Please address the concern about the automated pipeline in Weaknesses.
* From my perspective, it is not quite correct to compute FID in the editing setting, especially when there are only 92 real images. Could you please provide more details on this?

---

> ### Author Response · Authors · 2023-11-20
> **Response to Reviewer - Part 1**
>
> We thank the reviewer for providing constructive comments, appreciating our extensive study and findings!
>
> Below we provide responses to the reviewer's questions:
>
> **Only 6 out of 13 edits are supported.** : The reviewer rightly points out that our OWL-ViT pipeline is used to evaluate only 6 out of the 13 edit types. This is because these are the edit types for which the pipeline produces scores that are strongly correlated with human scores. For the remaining 7 edit types, the OWL-ViT pipeline, similar to CLIP produces scores that are only weakly correlated with human scores (see updated Sec. (K)). We did not want to throw the baby out with the bathwater, so we proposed the OWL-based pipeline as a proxy for a human study for the first 6 edit types. We recommend our standardized human template to evaluate the remaining 7 edit types which does not require using edited images from earlier methods to evaluate a given editing method. We note that this standardized template is a contribution in and of itself, and can be used by future editing methods to compare to earlier methods.
> While this does not present a ‘unified’ evaluation pipeline, we note that the OWL-based pipeline offers a more fine-grained metric compared to CLIP Scores (see Sec. (J)). .
>
> While we would have liked to also contribute an automated way of evaluating the remaining 7 edits types, we have left this for future work - perhaps a future extension of EditVal!
>
> We have also updated Sec. (5) in the paper to provide guidelines/a checklist for how to evaluate an image editing method on EditVal and how to interpret the results.
>
> **The pipeline can only check the object presence, location and size. For example, it cannot recognize if a cup stands naturally on the bench for “add a cup on the bench”. These aspects make this evaluator quite vulnerable.** : We recognize that our current pipeline has a limitation in determining whether a cup is naturally placed on a bench. We consider this issue as part of the larger challenge of assessing text-to-image generation methods across all conceivable scenarios. Our work represents a step forward in evaluating these methods within a specific subset of scenarios -  For e.g.,(i)  if a new object is added (object-addition), then our pipeline checks if both the old and the new object is present; (ii) If an object is altered (alter-parts), then the alteration is within that given object, leveraging predicted bounding boxes.
>
> We consider our automated evaluation pipeline to be a significant step from CLIP Scores, which are the current standard for evaluating text-to-image models and provide little fine-grained understanding of where models are failing and succeeding. As the evaluation of generative models continues to evolve, we anticipate that criteria addressing issues such as the natural placement of objects can be smoothly incorporated into future versions of our framework.
>
> **It does not evaluate image fidelity and image-context preservation. One still needs FID and DINO/CLIP scores which are not specifically designed for editing evaluation.** : The reviewer correctly notes that our pipeline uses FID and DINO scores to evaluate image fidelity and image-context preservation (which correspond to the second and third question in our human study). Even though these metrics are not specifically designed for editing evaluation, we find that they are strongly correlated with our human study scores (see Sec. (4.3) and Sec. (D) in the appendix). We have updated the text in Sec. (3) to emphasize that our automated pipeline uses FID/DINO metrics to check image fidelity and context preservation, so that there is no confusion.

---

> > ### Author Response · Authors · 2023-11-20
> > **Response to Reviewer - Part 2**
> >
> > **Probably, it would be reasonable to finetune some visual language model on the collected human scores and obtain something similar to ImageReward[1] but for the editing tasks. In addition, one can consider combining it with the proposed detector-based algorithms** : ImageReward [1] method focuses on collecting 9k samples annotated on image-quality as well as overall text-alignment, and utilizes this data for better finetuning of T2I diffusion models. However, as explored in the limitations of ImageReward [1], even these many samples may not be enough for learning an accurate reward model. To follow the same trajectory as ImageReward, we need extensive ground truth annotations on a much larger scale for each editing type and question type across all the given methods for finetuning a vision-language-based reward model. This necessitates us to have extensive human annotations with minimal annotation noise; further in the case of a novel editing type, we need to annotate and retrain the reward model again for this unseen editing type as well. Therefore, we defer this extension to future works.
> > The primary contribution of EditVal lies in proposing a new evaluation framework which integrates the evaluations of editing methods on multiple editing types of varying complexity. Any future editing method can easily be benchmarked on these editing types, and new editing types can be coded as directed in EditVal's open-source code.
> >
> > **From my perspective, it is not quite correct to compute FID in the editing setting, especially when there are only 92 real images. Could you please provide more details on this?** : We agree with the reviewer that computing FID score in case of less number of samples (such as 648 edited images in our case, where 92 is the unique number of original images) may not reflect the visual quality accurately and would be an underestimate of image quality. As cited in the original FID paper, it requires atleast 10k samples to accurately depict the generation quality; this may not be practically feasible. However, given that the same number of images are used to compute FID for each editing method, it can still be used for comparing different editing methods. We would also like to state that this critical flaw of the underestimated FID metric is eliminated if we use the DINO score as implemented in EditVal pipeline as the DINO score is independent of the number of samples. We have added a note on this in Sec. (4.3).

---

> > > ### Author Response · Authors · 2023-11-22
> > > **Looking Forward To Your Response!**
> > >
> > > We thank you again for your valuable feedback and comments which has helped to strengthen our paper. As the discussion period is ending soon, we would really appreciate if you could let us know if our responses have addressed your concerns. We will be happy to answer any further questions and address any remaining concerns to the best of our abilities in the remaining time!

---

> > > > ### Comment · Reviewer_mpAW · 2023-11-23
> > > > **Response to the rebuttal**
> > > >
> > > > I would like to thank the authors for their valuable clarifications and updates, which improved the initial submission. I'm glad that the FID problem is addressed and would suggest using individual sample estimators instead (ImageReward, PickScore or HPSv2), which are trained to correlate with human preference.
> > > >
> > > > Nevertheless, I still have concerns about the practical usefulness and novelty of the proposed automated pipeline. It essentially relies on the existing measures except for the detection-based one, which, in my humble opinion, brings limited value to the overall pipeline.
> > > >
> > > > I appreciate the authors' effort in thorough data collection and evaluation of the existing text-guided editing methods. However, I do not feel that this is enough for acceptance and hence tend to keep my score.

---

### Official Review · Reviewer_KGHZ · 2023-10-30

**Soundness:** 2 fair
**Presentation:** 1 poor
**Contribution:** 3 good
**Rating:** 5
**Confidence:** 4

**Summary:**

The paper proposes an evaluation protocol for text-guided image editing methods, and evaluates a set of 8 recent diffusion-based editing methods. Authors first build an editing benchmark using ChatGPT, comprising 92 images and 19 classes from the COCO dataset. Each image is associated with a set of pre-defined editing instructions linked to objects categories and manually defined editing types. This benchmark is then used to evaluate the 8 editing methods in two ways: 1) using a AMT user study, and 2) using hand crafted object detection-based rules for object centric editing tasks (e.g. object replacement or addition) to automatically evaluate editing success.  The performance of the methods is discussed according to these metrics, and the correlation between the human study and automated evaluation is investigated as well.

**Strengths:**

Image editing is a challenging task to evaluate, notably due to its subjective nature. Existing metrics and evaluation protocols are insufficient, despite the current high popularity of the topic. Therefore, authors are addressing an important and timely research topic. Having a systematic evaluation protocol for editing tasks can strongly benefit methodological development.

Authors carried out a large amount of work, manually curating images from the COCO dataset and designing a set of editing instructions. The detailed evaluation and analysis of 8 popular editing methods is particularly interesting, highlighting their strengths and limitations.

The idea of going beyond global image scores and leveraging object detection tasks is interesting and has potential to provide informative insights.

**Weaknesses:**

One big limitation of the paper is its poor presentation. It is very crowded, with a lot of vspace adjustments, making the reading experience uncomfortable. Several facts are repeated numerous times, notably editval’s description and main contributions, while key elements are left for the reader to find in the appendix (e.g. the choice of the COCO dataset as source of images).  The definition of editVal instead is not clear, in certain parts of the paper it is described as the data, edits and automated eval, and in others (e.g. the introduction) it includes the human study as well.

The paper is presenting many contributions : building a benchmark, analyzing pre-existing methods through a human study, developing an automated evaluation metric and comparing this metric’s performance to user preferences. It is impossible to address all thoroughly and clearly, leading to a lack of in depth discussion and exploration.  For example, the user study can provide a lot of insights over the different editing methods’ behaviours, but only the top 4 methods (selected according to an unknown criterion) are briefly discussed in 4.2. Comparing the performance of methods that require fine-tuning vs training free, or methods that use similar editing mechanisms, for example, could be very interesting. Perhaps the work could be more impactful if split in multiple papers or sent to a venue where more space is available.
Authors also overclaim their contributions, presenting their automated pipeline as an alternative to the human study. However, their detection based strategy can only evaluate a subset of editing tasks, and only provides a binary success/fail output. In addition, additional editing criteria (content preservation, image quality) cannot be measured this way, with authors reverting to standard metrics (FID, DINO scores) to complete their evaluation process.  It is not clear whether these two additional metrics are part of the automated evaluation pipeline.

The related work section is too limited. An important topic of the paper is the lack of reliable evaluation metrics for editing tasks, yet pre-existing works are not reviewed (besides CLIP). For examples, authors do not discuss recent techniques such as ImageReward (Xu et al, Neurips 2023) or Pickscore (Kirstain et al. , 2023). Furthermore, the detection based metric should be compared to these other metrics  in terms of human preference correlation, to highlight the advantages of the proposed technique.
It should also be noted that authors claim a strong correlation between their human study and their proposed metric, yet DINO scores show a much stronger correlation (fig 10), than the owl-vit based metric (fig. 6). This further highlights the importance of comparing the proposed technique to pre-existing works.

**Questions:**

-In section 3.1, it is mentioned that the 19 categories with the highest overlap across editing types were selected. What does overlap across editing types mean? What was the selection criterion?

-It is mentioned that chatGPT is used to generate a list of plausible editing changes. Are these quality controlled as well ?

-How did author select the top 4 performing methods to show in Figure 4? Was a global score computed, or a ranking across editing types and question type?

-Dreambooth and textual inversion were not initially developed for editing purposes, why include these methods?

-COCO is still a challenging dataset for object detection tasks. Was the dataset curated for simpler detection tasks? Does it occur that OWL-vit fails to detect the objects to edit?

---

> ### Author Response · Authors · 2023-11-20
> **Response to Reviewer - Part 1**
>
> We thank the reviewer for the constructive comments and appreciating the fact that we tackle a timely and important research topic!
>
> Below we provide responses to the reviewer's questions:
>
> **One big limitation of the paper is its poor presentation. It is very crowded, with a lot of vspace adjustments, making the reading experience uncomfortable. Several facts are repeated numerous times, notably editval’s description and main contributions, while key elements are left for the reader to find in the appendix (e.g. the choice of the COCO dataset as source of images)** :
> We apologize that the paper’s presentation felt crowded - we had many results to share! To address this, we have:
> (i) Updated Sec. (3) (especially 3.1) to be more readable with additional context to remove ambiguity.
> (ii) Removed the \vspace adjustments from all the sections to make the paper more presentable.
> (iii) Removed irrelevant information from 3.1 to the Appendix, to only highlight the main parts of the data-collection and annotation process.   (iv) We have also clarified the definition of EditVal in the introduction and other parts of the paper (Sec. 3 in particular). Specifically, we consider EditVal as a benchmark for evaluating text-guided editing methods. EditVal, therefore, includes all the pieces needed to evaluate a method - the data and its annotations, the automated evaluation pipeline (for 6 edit-types) and the templates for the standardized human-study (for 7 edit-types). (v) In addition, we have added a new section (see Sec. (5)) with guidelines on how to use it, to remove any confusion.
>
> We hope these changes collectively make the paper more readable and presentable!
>
> **The paper is presenting many contributions : building a benchmark, analyzing pre-existing methods through a human study, developing an automated evaluation metric and comparing this metric’s performance to user preferences. It is impossible to address all thoroughly and clearly, leading to a lack of in depth discussion and exploration. For example, the user study can provide ....** : We thank the reviewer for raising this point. We did consider splitting our contributions into multiple papers, but felt that we could not present a new benchmark without thoroughly validating it (through the automated and human-based assessment of pre-existing methods). We therefore relied on the appendix, with deeper explorations of the models and findings included there. To remedy this, we have moved some of the important points of our work to the main paper such as : (i) Better describing the dataset and its filtering process in updated Sec. (3.1); (ii) Reorganizing the presentation of results in Sec. (4.2). (iii) Provide guidelines on how to use EditVal in a new Sec. 5.
>
>
> In response to ‘overclaiming contribution that automated pipeline is an alternative to a human study’ -  In almost all cases, an automated pipeline (e.g. using OWL-ViT) cannot be used as a full replacement for a human study. As the reviewer notes, these pipelines often only give simplified outputs (e.g. binary success/fail) and cannot be used to evaluate more complex edit types (e.g. image quality). These simplified outputs however, can still provide useful signals for understanding model performance. So rather than throwing the baby out with the bathwater, EditVal leverages these for 6 of the edit types, and complements them with other metrics (e.g. FID, DINO) and a human study to evaluate the remaining 7 edit types.
>
> We consider these other metrics as part of the full automated pipeline, however, note that we have slightly conflated some terminology. We have updated the manuscript in Sec. (3.2.2)  to refer to the OWL-ViT based pipeline as the ‘detector-based automated pipeline’ and the DINO/FID metrics as the context-preservation pipeline. These two parts in conjunction make the automated pipeline in our paper.
>
> We show that our detector-based pipeline using OWL-ViT produces scores that are strongly correlated with human scores for the 6 edit types we use it to evaluate. OWL’s scores are poorly correlated with human scores for the remaining 7 edit types (similar to CLIP), which are more complex, hence we recommend the DINO metrics and human study templates for them. We have updated Sec. (K) for a discussion on this.
>
> We believe that EditVal takes a significant step beyond CLIP Scores, which are the current evaluation standard, to provide a more fine-grained way to evaluate image-editing methods leveraging our dataset and its annotations on different edit-types.

---

> > ### Author Response · Authors · 2023-11-20
> > **Response to Reviewer - Part 2**
> >
> > **The related work section is too limited. An important topic of the paper is the lack of reliable evaluation metrics for editing tasks, yet pre-existing works are not reviewed (besides CLIP). For examples, authors do not discuss recent techniques such as ImageReward (Xu et al, Neurips 2022...**: We thank the reviewer for bringing these works to our attention. During the project, we only consider published works and to our knowledge, ImageReward was still under review at NeurIPS ‘23 just before ICLR submissions. However, we have added and discussed these papers to our related works, and hope that future works will benchmark them against other methods on the EditVal benchmark.
> >
> > We note that ImageReward can evaluate the image editing quality of a method via  assessing the overall image quality and text alignment. However, our evaluation is multi-dimensional and granular in nature as we generate scores based on the  edit type for each edited image. Therefore, there is no straightforward way of comparing ImageReward and our automated detection pipeline scores; further to rank/compare the editing methods, a simple score averaging across edit-types would be a misleading comparison.
> >
> > In Sec. (J) - Appendix, we compare the detection-based metric to other metrics (e.g., CLIP-Score which is still widely used to evaluate text-guided image editing methods). We have added new results here, showing that the CLIP-Score metric has low correlation with the human-study annotations for spatial edits, therefore highlighting that OwL-ViT is indeed a better choice than CLIP-Score.
> >
> >
> > **It should also be noted that authors claim a strong correlation between their human study and their proposed metric, yet DINO scores show a much stronger correlation (fig 10), than the owl-vit based metric (fig. 6). This further highlights the importance of comparing the proposed technique to pre-existing works..**: The reviewer raises a relevant point. We note that Fig (6) compares OwL-ViT’s scores with human scores from only the first question in the human-study template. This question specifically asks if the edit is successful or not based on the text prompt. On the other hand, the correlations with DINO scores in Fig (10) should be compared with the third question in the human-study template which asks if the image context is preserved or not.
> >
> > Given that both these metrics are comparable to different parts of the human study, they are not directly comparable against each other. We have emphasized these points more in the updated version of our paper in Sec. (3.2.2).
> >
> > **-In section 3.1, it is mentioned that the 19 categories with the highest overlap across editing types were selected. What does overlap across editing types mean? What was the selection criterion?** : We apologize that these details were not clear in the paper. We employed ChatGPT to compile a list of classes from MS-COCO that are amenable to various edit types (e.g., object-addition / alter-parts amongst others). We then used a small-scale human study (see Sec.(H))  to validate the feasibility of these edit operations on the identified classes. The resulting list of filtered classes and associated edit types, however, was large – with many classes being amenable to only 1 or 2 edit operations. We therefore engaged human raters to further refine the classes to the subset that had the highest overlap between the 13 edit-types. This process resulted in the 19 selected classes spread across all the 13 edit-types.
> >
> > This refined and smaller set of classes means that a method can be evaluated faster and more computationally efficiently on EditVal. Most text-guided editing methods involve a lengthy fine-tuning process per image, which increases if there are more classes in the dataset (and more images). This filtered subset ensures that even fine-tuning based editing methods can be evaluated within a reasonable amount of time. We have included more detail of this selection process in Section 3.1 to enhance clarity and transparency.
> >
> > **-It is mentioned that chatGPT is used to generate a list of plausible editing changes. Are these quality controlled as well ?** : The reviewer raises an important point! We indeed performed a quality control on the answers given by ChatGPT.  We refer the reviewer to Sec. (H) in the Appendix which contains details on how the ChatGPT prompts were validated. In particular, we used human raters to filter ChatGPT’s suggestions to obtain the plausible editing changes. We also use a small-scale human study to validate the classes given by ChatGPT for edit operations (as described in the answer above). These steps ensure that our benchmark design is robust and does not consist of any unrealistic edit operations.

---

> > > ### Author Response · Authors · 2023-11-20
> > > **Response to Reviewer - Part 3**
> > >
> > > **How did author select the top 4 performing methods to show in Figure 4? Was a global score computed, or a ranking across editing types and question type?** : We select the top 4 performing methods by computing the median score for the first question type related to "editing quality" for each editing method and aggregate it across all the 13 editing types. Based on this score, we choose the top 4 editing methods and normalize the annotation score of each question ({0,1,2,3}) to a scale of 0-1 and represent our results in Figure 4. We are happy to answer, if the reviewer has any more questions on this!
> > >
> > > **Dreambooth and textual inversion were not initially developed for editing purposes, why include these methods?** : We agree with the reviewer that Dreambooth and Textual Inversion were not initially developed for editing purposes. However, both methods learn a unique identifier for a given object in a scene. All the edit operations defined in our dataset consist of operations on a particular object in a scene (e.g., adding a new object to an existing object, changing the spatial positioning of an existing object). Therefore if one learns the object identifier, then combining it with new prompts can be used to edit images.
> > > Interestingly, our study reveals that Dreambooth is competitive with other methods which were specifically curated for text-guided image-editing (see Fig. (5) and qualitative examples in Fig. (1)).
> > >
> > > **COCO is still a challenging dataset for object detection tasks. Was the dataset curated for simpler detection tasks? Does it occur that OWL-vit fails to detect the objects to edit?** : We agree with the reviewer that MS-COCO is indeed a challenging dataset for object detection tasks. We note, however, that OwL-ViT [1]  is one of the strongest open-vocabulary models for object detection on MS-COCO. We do acknowledge, however, that in low quality or complex images OwL-ViT will sometimes fail. To mitigate this, we ensured that the MS-COCO images we selected for EditVal were high quality and simple (e.g. less clutter in the background, only containing up to 2 objects based on the COCO annotations). With a confidence threshold of 0.1 (i.e., the same confidence threshold that we use for automatic evaluation), OwL-ViT archives 90.1% accuracy on the selected subset of data for the detection of the main objects in the images.  We describe this in Sec. (3) and Sec. (B.1) in the appendix, and also provide qualitative examples in Appendix (M).
> > >
> > >
> > > [1]. Matthias Minderer, Alexey Gritsenko, Austin Stone, Maxim Neumann, Dirk Weissenborn, Alexey Dosovitskiy, Aravindh Mahendran, Anurag Arnab, Mostafa Dehghani, Zhuoran Shen, Xiao Wang, Xiaohua Zhai, Thomas Kipf, and Neil Houlsby. Simple open-vocabulary object detection with vision transformers, 2022

---

> > > > ### Author Response · Authors · 2023-11-22
> > > > **Looking Forward To Your Response!**
> > > >
> > > > We thank you again for your valuable feedback and comments which has helped to strengthen our paper. As the discussion period is ending soon, we would really appreciate if you could let us know if our responses have addressed your concerns. We will be happy to answer any further questions and address any remaining concerns to the best of our abilities in the remaining time!

---

### Official Review · Reviewer_YjMj · 2023-11-01

**Soundness:** 3 good
**Presentation:** 3 good
**Contribution:** 3 good
**Rating:** 6
**Confidence:** 3

**Summary:**

The paper introduces a comprehensive benchmark specifically designed to evaluate text-guided image editing methods, effectively addressing a noticeable gap in the realm of image editing. This benchmark assembles a dataset encompassing 13 potential types of edits and proposes two evaluation pipelines, consisting of both an automated pipeline and a human-study template. The automated pipeline leverages vision-language models for the assessment of object-centric modifications, while the human-study template utilizes Amazon Mechanical Turk (AMT) to gather responses to a curated set of questions. The paper conducts evaluations on 8 state-of-the-art diffusion-based image editing methods, providing a valuable reference for future advancements in the field.

**Strengths:**

1. The paper tackles a critical and previously unaddressed issue in image editing, identifying the limitations inherent in existing benchmarks such as TedBench and EditBench.
2. A novel and holistic evaluation approach is introduced, incorporating both the automated method and the human study to assess a comprehensive range of 13 edit types.
3. The paper conducts thorough evaluations on eight of the latest image editing methods, serving as a good reference for future work.

**Weaknesses:**

1. There is a need for more detailed information regarding the implementation of the image editing methods. For example, methods such as Null-Text and SINE, are not based on instructions. It is confusing that instruction-based editing evaluations are applied to them.
2. The evaluation appears to lack focus on image-editing methods that deal with complex editing operations. For instance, methods like Diffusion Self-Guidance for Controllable Image Generation, which is designed to handle the shape, location, and appearance of objects, could be evaluated to show the performance of spatial manipulation.

**Questions:**

Please refer to weaknesses.

---

> ### Author Response · Authors · 2023-11-20
> **Response to Reviewer**
>
> We thank the reviewer for appreciating our work, especially the fact that we tackle a previously unaddressed issue in image editing evaluation and provide thorough evaluations!
>
> Below we provide our responses to the reviewer's questions:
>
> **There is a need for more detailed information regarding the implementation of the image editing methods. For example, methods such as Null-Text and SINE, are not based on instructions. It is confusing that instruction-based editing evaluations are applied to them** : We emphasize that, when utilizing Null-Text and SINE, we abstain from employing explicit instructions to generate edited images. We have provided details of our instruction-free prompts in Section I and Table 3-Appendix, and will aim to make these more prominent in the final manuscript. Specifically, the prompts are constructed by focusing on the object in the image (e.g., a bench) and its corresponding editing operation (e.g., object-addition). We ensure that the prompts share a similar nature across models, comprising the object and descriptive text indicating the editing operation (e.g., "A brown bench" when altering the bench color to 'brown'). However, there are instances where the prompt may be adjusted based on the specific editing method; for example, for Dreambooth, which necessitates a [V]* token, the curated prompt might be 'A brown [V]*bench'.
> In the case of instruction-based approaches like Instruct-Pix2Pix, we adhere to a generic instruction template. For instance, if the task involves adding a new object alongside an existing one, the prompt would be 'Add a <new-object> to the <object>'. These details are also updated in Sec. (I).
>
> **The evaluation appears to lack focus on image-editing methods that deal with complex editing operations. For instance, methods like Diffusion Self-Guidance for Controllable Image Generation, which is designed to handle the shape, location, and appearance of objects, could be evaluated to show the performance of spatial manipulation.** :  We thank the reviewer for bringing this paper to our attention. At the time of submission, this paper was still under review at NeurIPS ‘23, and so we did not consider it in our evaluation. During the rebuttal, we tried to evaluate their method on our EditVal benchmark. However, the authors have not yet released their code. Their project page (https://dave.ml/selfguidance/) and paper (https://arxiv.org/abs/2306.00986) does not contain any links for code which would enable us to evaluate this method on EditVal.  We have added a note on this in the updated version of our paper (see Sec. 2).
> However, we note that our benchmark is flexible to be used with any text-guided image-editing method in the future. Our human-study template as well as automatic evaluation does not require comparing edited images from earlier methods and only utilizes the current method in question. We hope that when their code is publicly available, EditVal can be used to benchmark its performance in a standardized way against other image-editing methods.
>
> In our paper, we only choose a few popular editing methods to provide a starting point for comparing editing methods on EditVal, but with the flexibility of EditVal, any new method can be added!

---

> > ### Author Response · Authors · 2023-11-22
> > **Looking Forward To Your Response!**
> >
> > We thank you again for your valuable feedback and comments which has helped to strengthen our paper. As the discussion period is ending soon, we would really appreciate if you could let us know if our responses have addressed your concerns. We will be happy to answer any further questions and address any remaining concerns to the best of our abilities in the remaining time!

---

### Author Response · Authors · 2023-11-20
**Global Response to Reviewers**

We thank all the reviewers for their constructive comments towards improving our paper. We note that all the reviewers appreciated that our work tackles an important and unaddressed issue in evaluating text-guided image editing methods. In addition – Reviewer YjMj appreciated our holistic evaluation approach and large-scale findings, Reviewer KGHZ appreciated our annotated dataset and findings from our paper, Reviewer mpAW founds our conclusions and findings valuable and Reviewer b2qM appreciated the general purpose nature of our benchmark.

During the rebuttal period, we have individually responded to the reviewer’s questions and have updated the paper with valuable comments from the reviewers. Below, we summarize the updates in the revised paper:

- Improved the introduction (Sec. 1)  to be better readable.
-  Added missing related works.
- Removed \vspace adjustments to improve readability.
- Improved Sec. 3 which describes our benchmark to be more precise about the different pieces of our benchmark.
- Modified Sec. 4.2 and Sec. 4.3 to improve readability and takeaways.
- Added a new Sec. 5, which provides guidelines on how EditVal can be used as a checklist for evaluating text-guided image editing methods.  Overall, we believe that rather than relying on a single score, EditVal gives multiple signals towards the evaluation of editing methods – which can therefore serve as a checklist before deploying any text-guided editing method.
- Updated Sec. (J) with CLIP’s correlation scores with our human-study.
- Updated Sec. (K) with a discussion on why existing vision-language models are difficult to use for the edit-types not considered in the automated evaluation.

---

### Meta-Review · Area_Chair_jP3p · 2023-12-10

**Metareview:**

Summary

The authors propose a benchmark protocol for text-baed image editing methods across the edit types as there has been no standard benchmark to compare image editing methods.

One common concern pointed out by the reviewers is the completeness of the benchmark method. To be more specific,
* Automated evaluation pipeline is only applicable to part of the edit types, leading to difficulty in scaling up
* Lack of capability to capture hallucination, image fidelity, and image-context preservation
* Lack of clarity, in-depth discussion, and exploration.

In the author-reviewer discussion, part of the concerns were addressed including image fidelity, image-context preservation evaluation issues. However, benchmarking is a very important contribution to the community; the correctness and the completeness of evaluation protocol should be treated with care to lead the research literature effectively. The reviewers still think it is not mature although they recognize the efforts and potential benefits.

**Justification For Why Not Higher Score:**

Benchmarking protocols are one of the most important contributions that serves as the standards in research direction. However, the reviewers are not convinced about the completeness and the correctness of the evaluation pipelines.

**Justification For Why Not Lower Score:**

N/A

---

### Decision · Program_Chairs · 2024-01-16

Reject